# Ultrafast simulation of large-scale neocortical microcircuitry with biophysically realistic neurons

Viktor J Oláh[1], Nigel P Pedersen[2], Matthew JM Rowan[1]*

[1]Department of Cell Biology, Emory University School of Medicine, Atlanta, United States; [2]Department of Neurology, Emory University School of Medicine, Atlanta, United States

**Abstract** Understanding the activity of the mammalian brain requires an integrative knowledge of circuits at distinct scales, ranging from ion channel gating to circuit connectomics. Computational models are regularly employed to understand how multiple parameters contribute synergistically to circuit behavior. However, traditional models of anatomically and biophysically realistic neurons are computationally demanding, especially when scaled to model local circuits. To overcome this limitation, we trained several artificial neural network (ANN) architectures to model the activity of realistic multicompartmental cortical neurons. We identified an ANN architecture that accurately predicted subthreshold activity and action potential firing. The ANN could correctly generalize to previously unobserved synaptic input, including in models containing nonlinear dendritic properties. When scaled, processing times were orders of magnitude faster compared with traditional approaches, allowing for rapid parameter-space mapping in a circuit model of Rett syndrome. Thus, we present a novel ANN approach allowing for rapid, detailed network experiments using inexpensive and commonly available computational resources.

*For correspondence:
mjrowan@emory.edu

## Editor's evaluation

This study describes the use of artificial neural network (ANN) methods to accurately replicate the biophysical behavior of detailed single-neuron models. The method has the potential to greatly increase the speed of neuronal modeling compared to conventional differential equation-based modeling, and scales particularly well for large network models. The authors demonstrate the fidelity of their ANN model cells over a wide range of stimulus and recording conditions including electrical and optical readouts.

## Introduction

Understanding the behavior of complex neural circuits like the human brain is one of the fundamental challenges of this century. Predicting mammalian circuit behavior is difficult due to several underlying mechanisms at distinct organizational levels, ranging from molecular-level interactions to large-scale connectomics. Computational modeling has become a cornerstone technique for deriving and testing new hypotheses about brain organization and function (*Sejnowski et al., 1988*; *Wolpert and Ghahramani, 2000*; *Dayan and Abbott, 2001*; *Kriegeskorte and Douglas, 2018*). In little more than 60 years, our mechanistic understanding of neural function has evolved from describing action potential (AP)-related ion channel gating (*Hodgkin and Huxley, 1952*) to constructing models that can simulate the activity of whole-brain regions (*Traub et al., 2005*; *Yu et al., 2013*; *Neymotin et al., 2016b*; *Chavlis et al., 2017*; *Turi et al., 2019*). Although tremendous advancements have been made

in the development of computational resources, the lack of available or affordable hardware for neural simulations currently represents a significant barrier to entry for most neuroscientists and renders many questions intractable. This is particularly well illustrated by large-scale neural circuit simulations. In contrast to detailed single-cell models, which have been a regular occurrence in publications since the 1990s (*De Schutter and Bower, 1994*; *Mainen et al., 1995*; *Migliore et al., 1995*; *Mainen and Sejnowski, 1996*; *Destexhe et al., 1998*; *Stuart and Spruston, 1998*; *Aradi and Holmes, 1999*; *Migliore et al., 1999*), parallel simulation of thousands, or even hundreds of thousands of detailed neurons have only become a possibility with the advent of supercomputers (*Markram et al., 2015*; *Bezaire et al., 2016*; *Arkhipov et al., 2018*; *Joglekar et al., 2018*; *Schmidt et al., 2018*; *Antolík et al., 2019*; *Schwalger and Chizhov, 2019*; *Billeh et al., 2020*). As these resources are still not widely accessible, several attempts have been made to mitigate the immense computational load of large-scale neural simulations by judicious simplification (*Wang and Buzsáki, 1996*; *Bartos et al., 2002*; *Santhakumar et al., 2005*; *Eppler, 2008*, *Cutsuridis et al., 2010*; *Nowotny et al., 2014*; *Bezaire et al., 2016*; *Yavuz et al., 2016*; *Teeter et al., 2018*; *Amsalem et al., 2020*; *Knight et al., 2021*; *Knight and Nowotny, 2021*, *Wybo et al., 2021*). However, simplification inevitably results in feature or information loss, such as sacrificing multicompartmental information for simulation speed (*Wang and Buzsáki, 1996*; *Bartos et al., 2002*; *Santhakumar et al., 2005*; *Bezaire et al., 2016*). Thus, there is a critical need for new approaches to enable efficient large-scale neural circuit simulations on widely available computational resources without surrendering biologically relevant information.

To counteract the increasing computational burden of ever-growing datasets on more traditional models, many fields have recently adopted various machine learning algorithms (*Sharma et al., 2011*; *Montavon et al., 2013*; *Meredig et al., 2014*; *Merembayev et al., 2018*; *Schütt et al., 2020*). Specifically, artificial neural networks (ANNs) are superior to conventional model systems both in terms of speed and accuracy when dealing with complex systems such as those governing global financial markets or weather patterns (*Holmstrom, 2016*; *Ghoddusi et al., 2019*). Due to their accelerated processing speed, ANNs are ideal candidates for modeling large-scale biological systems. The idea that individual neural cells could be represented by ANNs was proposed almost two decades ago (*Poirazi et al., 2003*); however, current ANN solutions are still unfit to replace traditional modeling systems as they cannot generate gradational neuronal dynamics needed for network simulations. Therefore, we aimed to develop an ANN that can (1) accurately replicate various features of biophysically detailed neuron models, (2) efficiently generalize for previously unobserved input conditions, and (3) significantly accelerate large-scale network simulations.

Here, we investigated the ability of several ANN architectures to represent membrane potential dynamics, in both simplified point neurons and multicompartment neurons. Among the selected ANNs, we found that a convolutional recurrent architecture can accurately simulate both subthreshold and suprathreshold voltage dynamics. Furthermore, this ANN could generalize to a wide range of input conditions and reproduce neuronal features following different input patterns beyond membrane potential responses, such as ionic current waveforms. Next, we demonstrated that this ANN could also accurately predict multicompartmental information by fitting this architecture to a biophysically detailed layer 5 (L5) pyramidal cell (PC; *Hallermann et al., 2012*) model. Importantly, we found that ANN representations could drastically accelerate large network simulations, as demonstrated by network parameter space mapping of a cortical L5 recurrent microcircuit model of Rett syndrome, a neurodegenerative disorder associated with cortical dysfunction and seizures (*Hagberg et al., 1985*; *Armstrong, 2005*, *Glaze, 2005*; *Chahrour and Zoghbi, 2007*). Thus, we provide a detailed description of an ANN architecture suitable for large-scale simulations of anatomically and biophysically complex neurons, applicable to human disease modeling. Most importantly, our ANN simulations are accelerated to the point where detailed network experiments can now be carried out using inexpensive, readily available computational resources.

## Results

To create a deep learning platform capable of accurately representing the full dynamic membrane potential range of neuronal cells, we focused on model systems proven to be suitable for multivariate time-series forecasting (MTSF). To compare the ability of different ANNs to reproduce the activity of an excitable cell, we designed five distinct architectures (*Figure 1*). The first two models were a simple linear model with one hidden layer (linear model, *Figure 1A*, blue) and a similar model equipped with

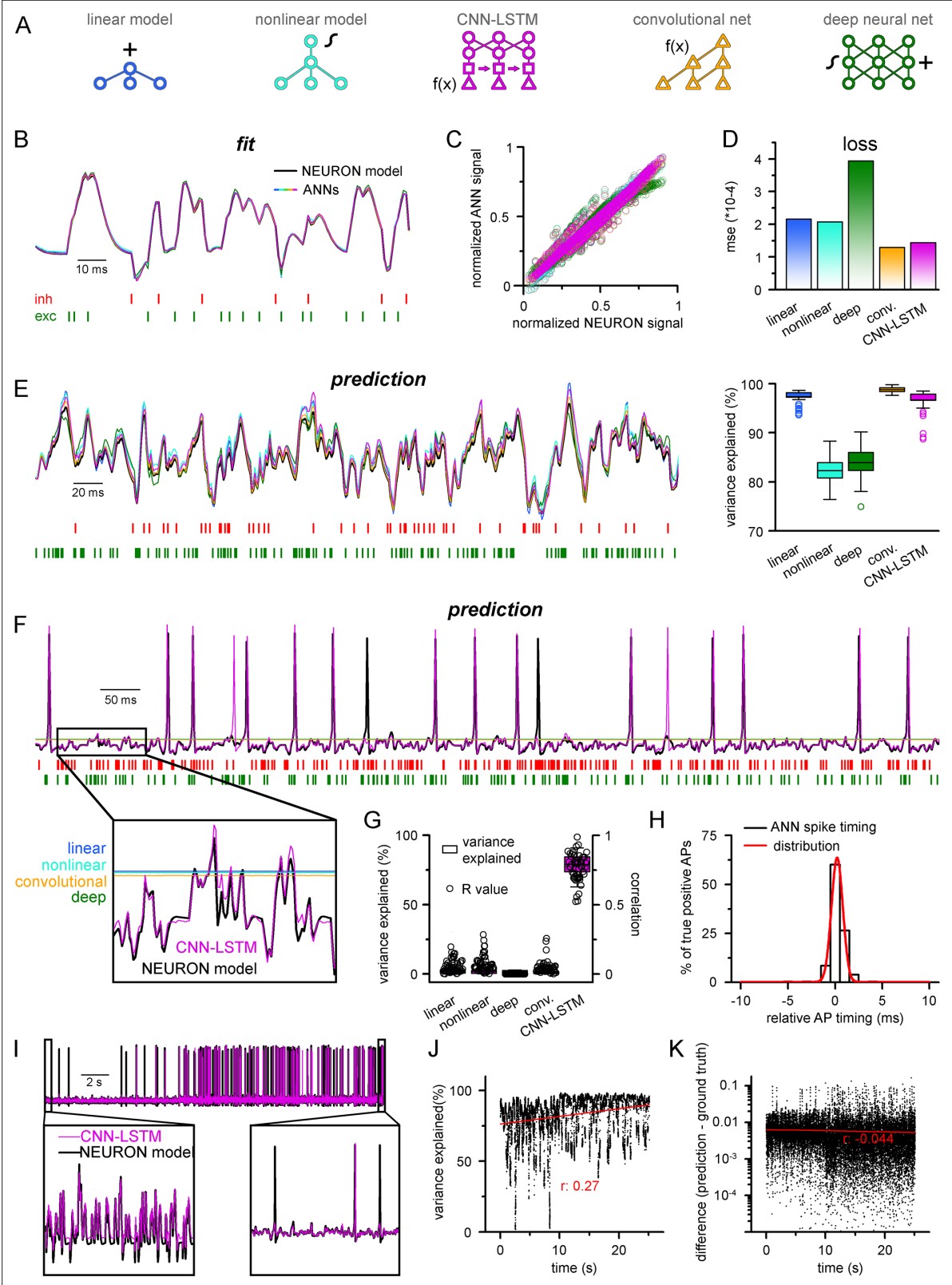

**Figure 1.** Single-compartmental neuronal simulations using artificial neural networks (ANNs).

(**A**) Representative diagrams of the tested architectures, outlining the ordering of the specific functional blocks of the ANNs. (**B**) Continuous representative trace of a point-by-point fit of passive membrane potential. (**C**) Point-by-point fit plotted against ground truth data (n = 45,000). (**D**) Mean squared error of ANN fits corresponds to the entire training dataset (n = 2.64 * 10⁶ datapoints). Single quantal inputs arrive stochastically with a

*Figure 1 continued on next page*

*Figure 1 continued*

fixed quantal size: 2.5 nS for excitatory, 8 nS for inhibitory inputs, sampling is 1 kHz. Red and green bars below membrane potential traces denote the arrival of inhibitory and excitatory events, respectively. (**E**) Representative trace of a continuous passive membrane potential prediction (left) created by relying on past model predictions. Explained variance (right) was calculated from 500-ms-long continuous predictions (n = 50). (**F**) Representative active membrane potential prediction by ANNs. (**G**) Explained variance (box chart) and Pearson's r (circles) of model predictions and ground truth data for the five ANNs from 50 continuous predictions, 500 ms long each. (**H**) Spike timing of the convolutional neural network-long short-term memory (CNN-LSTM) model calculated from the same dataset as panel (**G**). Color coding is the same as in panel (**A**). (**I**) Representative continuous, 25-s-long simulation of subthreshold and spiking activity. (**J**) Explained variance as a function of time during the 25-s-long simulation depicted in panel (**I**). Red line and r-value correspond to the best linear fit. (**K**) Difference between voltage traces produced by NEURON and ANN simulations. Red line and r-value correspond to the best linear fit.

The online version of this article includes the following figure supplement(s) for figure 1:

**Figure supplement 1.** Convolutional neural network-long short-term memory (CNN-LSTM) architecture for time-series forecasting.

nonlinear processing (nonlinear model, *Figure 1A*, cyan), as even relatively simple model architectures can explain the majority of subthreshold membrane potential variance (*Ujfalussy et al., 2018*). The third and fourth models consist of recently constructed time-series forecasting architectures, including a recurrent ANN (convolutional neural network-long short-term memory [CNN-LSTM], *Figure 1A*, magenta) consisting of convolutional layers (*Collobert and Weston, 2008*), long short-term memory (LSTM; *Hochreiter and Schmidhuber, 1997*; *Donahue et al., 2015*) layers, and fully connected layers, termed the CNN-LSTM network (*Figure 1—figure supplement 1*, *Shi, 2015*), and a more recently developed architecture relying on dilated temporal convolutions (convolutional net, *Figure 1A*, orange) (based on the WaveNet architecture; *Oord, 2016*; *Beniaguev et al., 2021*), which is superior to the CNN-LSTM in several MTSF tasks. The CNN-LSTM has the distinct advantage of having almost two orders of magnitude more adjustable parameters compared to the aforementioned ANNs. Finally, we selected a fifth architecture (deep neural net, *Figure 1A*, green) with a comparable number of free parameters to the CNN-LSTM, composed of 10 hidden layers, which operates solely on linear and nonlinear transformations. Before moving to neural cell data, each of the five selected architectures were evaluated using a well-curated weather time-series dataset (see 'Methods'). Each model performed similarly (0.070/0.069, 0.059/0.06, 0.089/0.094, 0.07/0.069, 0.092/0.095, mean absolute error on the validation/testing datasets for linear, nonlinear, convolutional net and CNN-LSTM, deep neural net architectures, respectively), demonstrating their suitability for MTSF problems.

## Prediction of point neuron membrane potential dynamics by ANNs

To test the ability of the five ANNs to represent input–output transformations of a neural cell, we next fitted these architectures with data from passive responses of a single-compartmental point-neuron model (NEURON simulation environment; *Hines and Carnevale, 1997*) using the standard backpropagation learning algorithm for ANNs (*Rumelhart et al., 1986*). Each model was tasked with predicting a single-membrane potential value based on 64 ms (a time window that yielded the best results both in terms of speed and accuracy) of preceding membrane potentials and synaptic inputs (*Figure 1A*). ANN fitting and query were run on a single-core central processing unit (CPU). We found that both the linear and nonlinear models predicted subsequent membrane potential values with low error rates (*Figure 1B*) with similar behavior in both the CNN-LSTM and convolutional architectures ($2.16 * 10^{-4} \pm 1.18 * 10^{-3}$, $2.07 * 10^{-4} \pm 1.11 * 10^{-3}$, $1.43 * 10^{-4} \pm 9.31 * 10^{-4}$, $1.29 * 10^{-4} \pm 9.42 * 10^{-4}$ mean error for linear, nonlinear, CNN-LSTM, and convolutional models, respectively). However, the deep neural network performed considerably worse than all other tested models ($3.94 * 10^{-4} \pm 1.56 * 10^{-3}$ mean error), potentially due to the nonlinear correspondence of its predicted values to the ground truth data (*Figure 1C and D*).

Next, we tested ANNs in simulation conditions similar to the traditional models. To this end, we initialized ANNs with ground truth data followed by a continuous query period in which forecasted membrane potential values were fed back to the ANNs to observe continuous unconstrained predictions. As expected from the fit error rates of single-membrane potential forecasting (*Figure 1D*), continuous predictions of the linear, convolutional, and CNN-LSTM models could explain the ground truth signal variance at high accuracy. At the same time, the deep neural net performed slightly worse (*Figure 1E*, 97.1 ± 1.2, 99.3 ± 1.4, 97.2 ± 2.2, 84.0 ± 3.2 variance explained for linear, convolutional, CNN-LSTM, and deep neural net architectures, respectively, n = 50). Surprisingly, the nonlinear model

produced the worst prediction for passive membrane potential traces (0.82 ± 0.03 variance explained, n = 50) despite performing the best on the benchmark dataset. Together, these results indicate that even simple linear ANNs can capture subthreshold membrane potential behavior accurately (*Ujfalussy et al., 2018*).

Next, we tested how these models perform on the full dynamic range of neural cells, which due to AP firing (which can also be viewed as highly relevant outlier data points) constitutes a non-normally distributed and thus demanding dataset for ANNs. Interestingly, we found that only the CNN-LSTM architecture could precisely reproduce both subthreshold membrane potential dynamics and spiking activity, while all other tested ANNs converged to the mean of the training dataset (*Figure 1F and G*, 4.4 ± 7.2%, 4.1 ± 6.9%, 0.5 ± 3.9%, 78.9 ± 6.7%, 4.4 ± 2.8% variance explained for linear, nonlinear, convolutional net and CNN-LSTM, deep neural net architectures, respectively, n = 50). We found that although the CNN-LSTM model explained substantially less variance for the active membrane potential traces (*Figure 1G*) than for subthreshold voltages alone (*Figure 1E*), the predictions showed high linear correlation with the ground truth signals (Pearson's $r$ = 0.76793 ± 0.10003, n = 50). For the four remaining ANN architectures, it is unlikely that convergence to the mean is caused by settling in local minima on the fitting error surface as ANNs have a large number of free parameters (2.07 * 10$^4$, 2.07 * 10$^4$, 2.47 * 10$^6$, 3.64 * 10$^5$, 1.95 * 10$^6$ free parameters for linear, nonlinear, deep, convolutional ANNs, and CNN-LSTM, respectively). Therefore, the chance of having a zero derivative for each parameter at the same point is extremely low (*Kawaguchi, 2016*), suggesting that erroneous fitting is the consequence of the limitations of these ANN architectures. Consequently, of the tested ANN architectures, the CNN-LSTM is the only model that could depict the full dynamic range of a biophysical neural model.

Closer inspection of the timing of the predicted APs revealed that the CNN-LSTM models correctly learned thresholding as the occurrence of the APs matched the timing of the testing dataset (*Figure 1H*; 83.94 ± 16.89% precision and 90.94 ± 12.13% recall, 0.24 ± 0.79 ms temporal shift for true-positive spikes compared to ground truth, n = 283), thus CNN-LSTM predictions yielded voltage traces with good initial agreement to NEURON signals. To test the long-term stability of these predictions, we next performed a longer (25 s) ANN simulation (*Figure 1I*). During this extended simulation, we did not observe significant deviation from the ground truth signal in terms of explained variance (*Figure 1J*) or absolute difference (*Figure 1K*) and these metrics even improved slightly. Taken together, we developed an ANN architecture that is ideally suited for predicting both subthreshold membrane potential fluctuations and the precise timing of APs on a millisecond timescale.

## Generalization of the CNN-LSTM architecture

To test the applicability of the CNN-LSTM for predicting physiological cellular behavior, we assessed the generalization capability of the architecture built for active behavior prediction (*Figure 1F*). Generalization is the ability of an ANN to respond accurately to novel data (*Hassoun, 1995*; *Graupe, 2013*). According to our hypothesis, if the CNN-LSTM correctly learned the mechanistic operations of a neural cell, then the architecture should behave appropriately when tasked with responding to novel quantal amplitudes and input patterns.

We first challenged the CNN-LSTM by administering excitatory inputs with variable quantal sizes (0.1–3.5 nS, 0.1 nS increment). Similar to the control NEURON model, the CNN-LSTM responded linearly in subthreshold voltage regimes (*Figure 2A*, Pearson's $r$ = 0.99, n = 35) and elicited an AP after reaching threshold. Independent evaluation of the NEURON model control revealed a surprisingly similar I/V relationship for the same quantal inputs (intercept, –0.003 ± 8.53 and –0.003 ± 0.001; slope for subthreshold linear I/V, 22.2 ± 0.41 and 23.31 ± 0.62; CNN-LSTM and NEURON model, respectively) and similar AP threshold (–58.03 mV and –56.64 mV for CNN-LSTM and NEURON model, respectively). Next, we tested temporal summation of excitatory inputs (*Figure 2B*). We found that the independently simulated NEURON model displayed similar temporal summation patterns to the CNN-LSTM for both sub- and suprathreshold events (*Figure 2B*). Finally, we combined the previous two tests and delivered unique temporal patterns of synaptic inputs with variable synaptic conductances randomly chosen from a normal distribution (mean: 2.5 nS; variance: 0.001 nS, *Figure 2C*). Again, the predictions of the CNN-LSTM architecture closely matched traces obtained from the NEURON model (*Figure 2D*, Pearson's $r$ = 0.81, n = 5000 ms) and the timing of the majority of the APs agreed with the ground truth data (91.02 ± 16.03% recall and 69.38 ± 22.43% precision, n = 50).

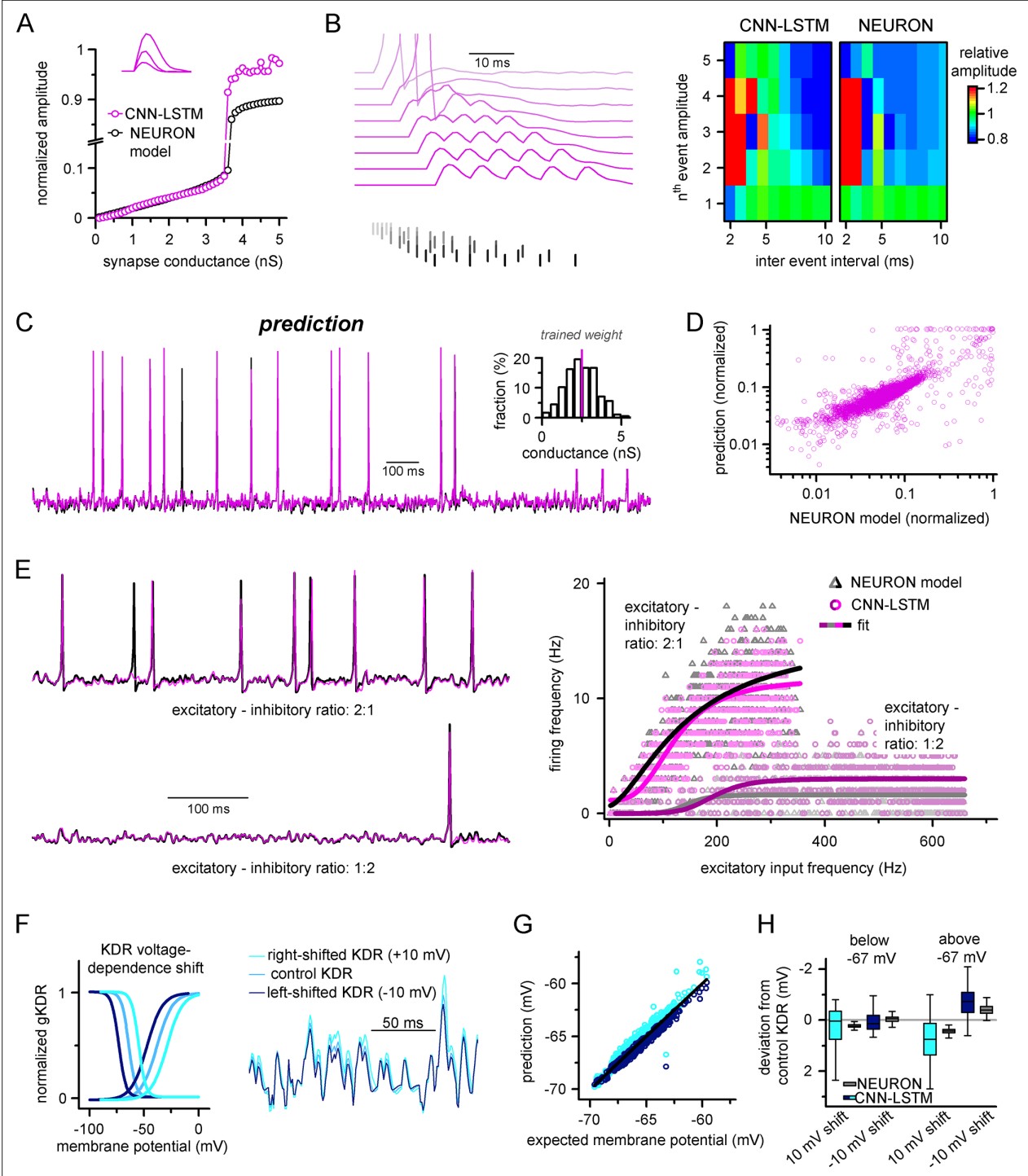

**Figure 2.** Ideal generalization of the convolutional neural network-long short-term memory (CNN-LSTM). (**A**) CNN-LSTM models predict similar subthreshold event amplitudes and action potential threshold (break in y-axis) for increasing input weight, compared to NEURON models. (**B**) CNN-LSTM models correctly represent temporal summation of synaptic events. Representative traces for different inter-event intervals (range: 2–10 ms, 1 ms increment) on the left, comparison of individual events in a stimulus train, relative to the amplitude of unitary events on the right. (**C**) Single-simulated active membrane potential trace in CNN-LSTM (purple) and NEURON (black) with variable synaptic input weights (left). The inset shows the distribution of synaptic weights used for testing generalization, with the original trained synaptic weight in purple. CNN-LSTM predicted membrane potential values plotted against NEURON model ground truth (right). Plotted values correspond to continuously predicted CNN-LSTM traces. (**D**) CNN-LSTM model predictions are accurate in various synaptic environments. Firing frequency was quantified upon two different excitation–inhibition ratios (2:1 – representative top trace on the left and bright magenta circles on the right, 1:2 – representative bottom trace on the left and dark magenta circles on

*Figure 2 continued on next page*

*Figure 2 continued*

the right). (**E**) Subthreshold effects of potassium conductance biophysical alterations are correctly depicted by the CNN-LSTM. Voltage dependence of the delayed rectifier conductances is illustrated on the left and their effect on subthreshold membrane potential is shown on the right (control conditions are shown in blue, 10 mV left-shifted delayed rectifier conditions in navy blue and 10 mV right-shifted conditions are shown in teal). (**F**) CNN-LSTM membrane potential predictions for left- (navy) or right-shifted potassium conditions are compared to control conditions., Membrane potential responses below and above –67 mV are quantified for the two altered potassium conductances in NEURON simulation and CNN-LSTM predictions. The effects of biophysical changes of potassium channels were only apparent at membrane potentials above their activation threshold (–67 mV). (**G**) Artificial neural networks (ANNs) fitting NEURON models with left-shifted (dark blue) and right-shifted (light blue) KDR conductances are plotted against membrane potential responses of ANNs with control KDR conductances. The separation of the two responses shows voltage response modulation of KDR at subthreshold membrane potentials. (**H**) Membrane potential responses of NEURON and ANN models below and above resting membrane potential (–67 mV).

In the initial training dataset for the CNN-LSTM, the ratio of excitatory and inhibitory events (8:3) was preserved while the total number of synaptic inputs was varied. We noticed that the firing rate of this model did not scale linearly with the number of synapses as initially expected in the presence of inhibitory inputs (*Enoki et al., 2001*). Thus, we systematically mapped AP firing of model cells in two different excitatory-inhibitory ratios with at a wide range of synaptic input frequencies (*Figure 2E*). We noted that varying excitation and inhibition could interact with each other in various ways, creating arithmetic operations like subtraction, division, or normalization (*Carandini and Heeger, 2011*). We approximated the resulting firing rates with two different models (see 'Methods'). We found that the logistic function representing divisive normalization best fit our results (*Bhatia et al., 2019*) (Akaike information criterion [AIC] for linear models representing subtractive and divisive inhibition versus AIC for logistic function: 983.3 ± 231.66 and 905.87 ± 200.92, respectively, n = 700 each). Notably, the CNN-LSTM model was able to replicate firing responses to these variable synaptic conditions ($R^2$ values when comparing logistic fits for NEURON and CNN-LSTM models in 2:1 excitation–inhibition ratio: 0.996, for 1:2 excitation–inhibition ratio: 0.9, n = 700), further demonstrating the ability of the neuronal net to reproduce key features of neuronal excitability without prior entrainment.

Due to the opaque nature of neural net operations (*Castelvecchi, 2016*), it is reasonable to assume that instant modification of the trained architecture to account for specific biophysical alterations may not be feasible, highlighting a potentially significant shortcoming of our approach. However, the complexity of encoded features is correlated with the depth of the encoding layer in hierarchically constructed neural networks (*Egmont-Petersen et al., 2002*), which can be exploited through partial retraining. To test whether the ANN could accurately handle a specific biophysical change, we constructed a simple NEURON model equipped with a delayed rectifier $K^+$ conductance with variable voltage dependences (*Oláh et al., 2021*; *Figure 2F*). Nonlinear signal summation at different subthreshold voltages was noted after shifting the steady-state activation and inactivation of the $K^+$ conductance (*Figure 2F*). From this model, a single CNN-LSTM model was fitted to the control $K^+$ condition. Subsequently, the CNN-LSTM model layers were frozen, with the exception of the (upper) fully connected layers, which were trained for 10 min on NEURON traces with either a 10 mV leftward or rightward shift introduced to the voltage dependence of the potassium conductance. All three models were in good agreement with the NEURON simulation results and provided similar deviations in subthreshold membrane potential regimes compared to control conditions (*Figure 2G and H*, below resting membrane potential: –0.13 ± 0.36, –0.11 ± 0.03, –0.01 ± 0.23, 0.01 ± 0.06; above resting membrane potential: –0.4 ± 0.43, –0.22 ± 0.55, 0.35 ± 0.27, 0.2 ± 0.1 for CNN-LSTM right-shift, NEURON right-shift, CNN-LSTM left-shift, and NEURON left-shift, respectively, n = 270), indicating that CNN-LSTM can be rapidly adapted to account for biophysical alterations.

NEURON models can calculate and display several features of neuronal behavior in addition to membrane potential, including ionic current flux. To test how our CNN-LSTMs perform in predicting ionic current changes, we supplemented ANN inputs with sodium ($I_{Na}$) and potassium currents ($I_K$) and tasked the models to predict these values as well. The accuracy of the CNN-LSTM prediction for these ionic currents was similar to membrane potential predictions (*Figure 3*, Pearson's r = 0.999 and 0.99 for fitting, n = 5000, variance explained: 15.1 ± 11.6% and 82 ± 6.1%; prediction correlation coefficient: 0.85 ± 0.08 and 0.81 ± 0.1, n = 5, for $I_K$ and $I_{Na}$, respectively) while the other ANNs again regressed to the mean.

Finally, we explored whether ANNs could represent nonlinear synaptic responses. Thus, we constructed single-compartmental models with two-component AMPA-NMDA containing synapses

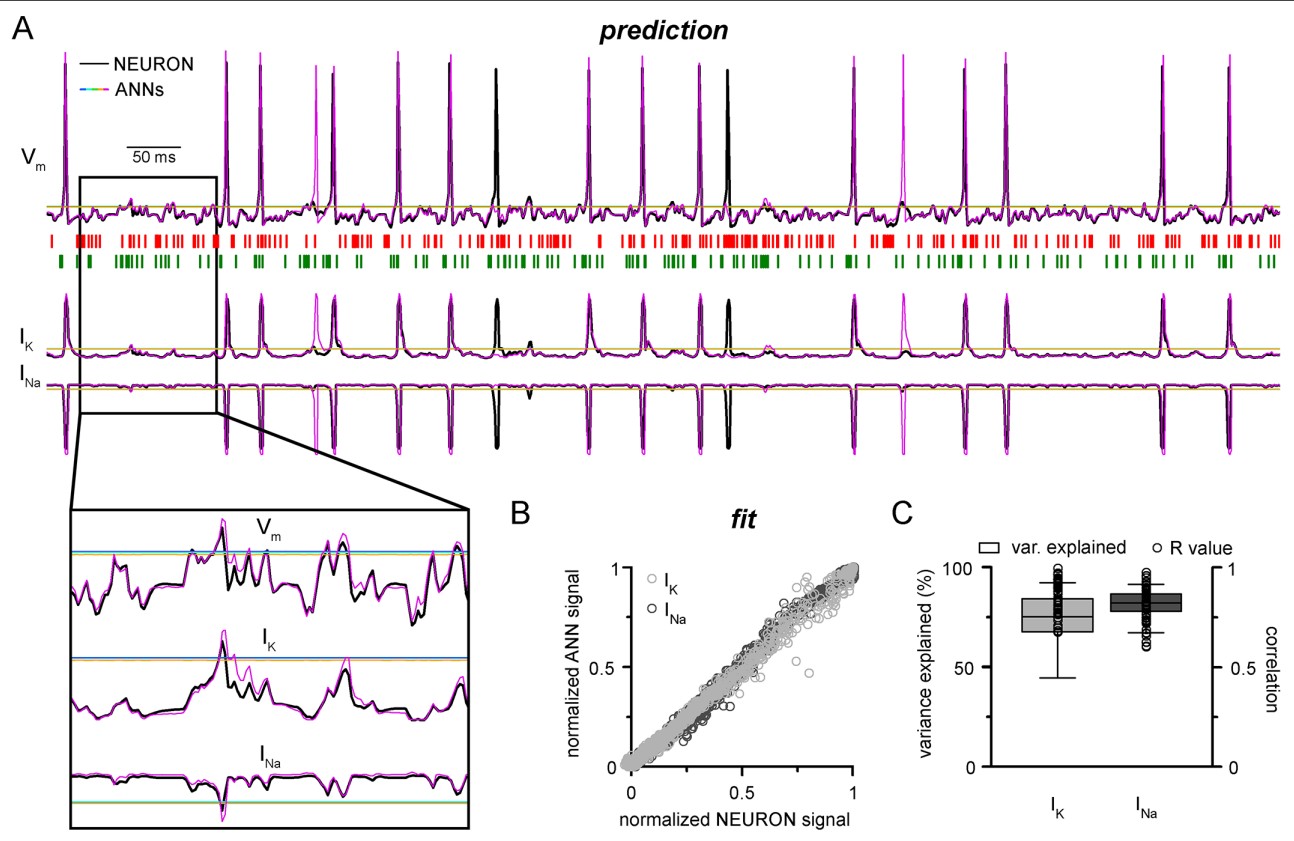

**Figure 3.** Convolutional neural network-long short-term memory (CNN-LSTM) prediction of neuronal mechanisms beyond somatic membrane potential. (**A**) Representative membrane potential ($V_m$, top) and ionic current ($I_K$, potassium current; $I_{na}$, sodium current, bottom) dynamics prediction upon arriving excitatory (green, middle) and inhibitory (red, middle) events. Enlarged trace shows subthreshold voltage and current predictions. Color coding is same as for *Figure 1*. (black, NEURON model traces; magenta, CNN-LSMT; blue, linear model; teal, nonlinear model; green, deep neural net; orange, convolutional net). Notice the smooth vertical line corresponding to predictions by artificial neural networks (ANNs), with the exception of CNN-LSTM. On bottom left, magnified view illustrates the subthreshold correspondence of membrane potential and ionic current traces. (**B**) CNN-LSTM models accurately predict ionic current dynamics. Normalized ANN predictions are plotted against normalized neuron signals for sodium (dark gray, left) and potassium currents (light gray). (**C**) Variance of suprathreshold traces is largely explained by CNN-LSTM predictions (right, color coding is same as in panel [**B**], left). Correlation coefficients are superimposed in black.

and inhibitory synapses. AMPA-NMDA model responses were voltage-dependent and produced nonlinear response curves with respect to AMPA alone (*Figure 4A*). Importantly, the CNN-LSTM architecture recreated the nonlinear response amplitude and time-course characteristic of AMPA-NMDA synapse activation (*Schiller et al., 2000*; *Major et al., 2008*; *Branco and Häusser, 2011*; *Kumar et al., 2018*).

A well-defined functional role of NMDA receptors is coincidence detection, which allows boosting of consecutive subthreshold signals well beyond passive integration (*Takahashi and Magee, 2009*; *Shai et al., 2015*). To test whether our ANN could reliably perform coincidence detection, we simulated two excitatory inputs in NMDA-AMPA or AMPA-alone models. Closely spaced stimuli could generate significantly boosted EPSPs in models with NMDA-AMPA (*Figure 4B*). We found that both NEURON and ANN models exhibited strongly boosted excitatory signals within a well-defined ISI time window (±12 ms) when NMDA-AMPA receptors were activated, which could produce APs. Under physiological conditions, NMDA receptors have been reported to critically influence the AP output of neuronal cells (*Smith et al., 2013*). Thus, we subjected NEURON models to a barrage of excitatory and inhibitory inputs, such that AP generation was limited in the absence of NMDA (*Figure 4C*). Adding NMDA resulted in increased spike output (*Figure 4C*). Across several NMDA conditions, output in the NEURON and ANN models was indistinguishable (*Figure 4C*, 12.42 ± 1.36 and 12.39 ± 2.3 Hz firing, respectively, in condition where 100% synapses contained NMDA receptors, n = 50).

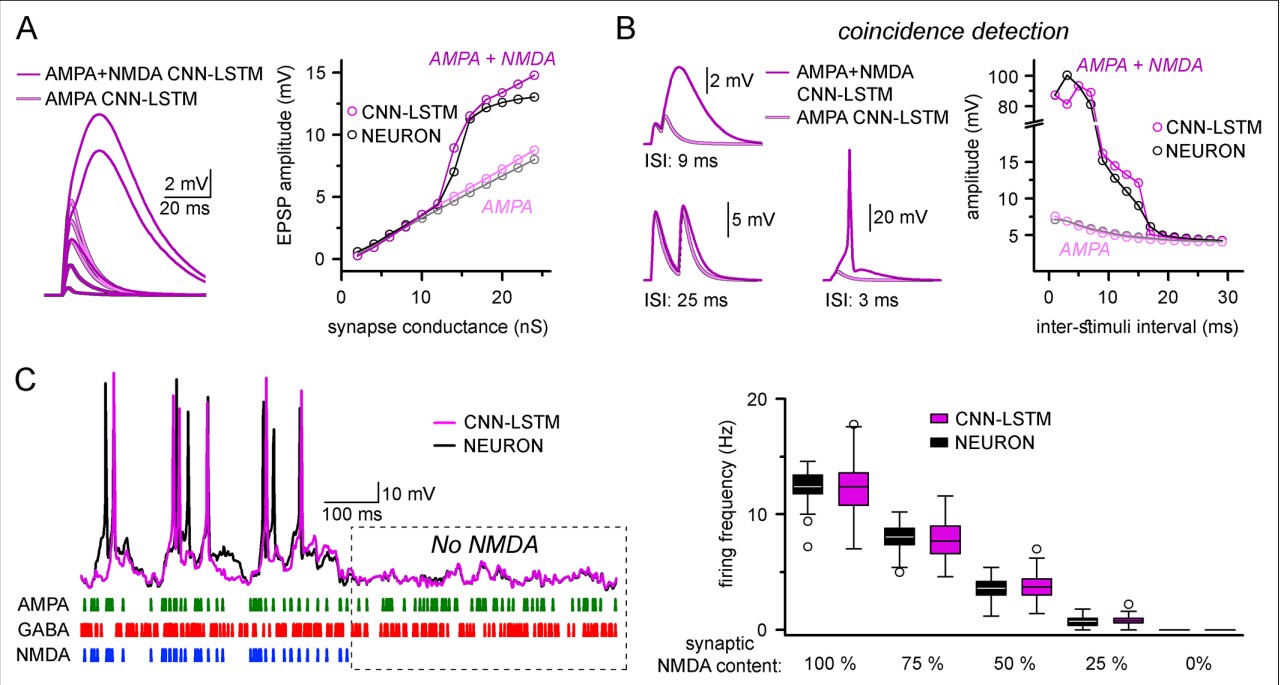

**Figure 4.** Accurate representation of nonlinear synaptic activation by convolutional neural network-long short-term memory (CNN-LSTM). (**A**) Representative synaptic responses with variable synaptic activation, CNN-simulated AMPA receptors (light magenta) and AMPA + NMDA receptors (magenta), on the left. AMPA + NMDA response amplitudes nonlinearly depend on the activated synaptic conductance (magenta, CNN-LSTM; black, NEURON), compared to AMPA responses (light magenta, CNN-LSTM; gray, NEURON), on the right. (**B**) NMDA response nonlinearity enables coincidence detection in a narrow time window, resulting in action potential (AP) generation at short stimulus intervals. (**C**). Neuronal output modulation is dependent on synaptic NMDA receptor content in a naturalistic network condition. Representative traces on the left (CNN-LSTM, magenta; NEURON, black). Summary depiction of firing frequencies with varying amounts of NMDA receptor activation (percentages denominate the synaptic NMDA-AMPA fraction).

Together, these results demonstrate that the CNN-LSTM correctly learned several highly specialized aspects of neuronal behavior.

## Predicting the activity of morphologically realistic neurons using ANNs

Neurons multiply their adaptive properties by segregating different conductances into separate subcellular compartments (*Magee and Cook, 2000*; *Kole et al., 2008*; *Losonczy et al., 2008*; *Kim et al., 2012*; *Rowan et al., 2014*; *Stuart and Spruston, 2015*; *Brunner and Szabadics, 2016*; *Stuart et al., 2016*). Thus, in addition to simplified input integrating point neurons, a substantial portion of neuronal models developed in recent decades intended to address subcellular signal processing via detailed multicompartmental biophysical cellular representations (*Major et al., 1994*; *Mainen and Sejnowski, 1996*; *Vetter et al., 2001*; *Hallermann et al., 2012*; *Brunner and Szabadics, 2016*; *Oláh et al., 2020*). Therefore, our next aim was to examine how well ANNs describe multicompartmental information. To this end, a training dataset of synaptic inputs and corresponding somatic voltage responses was generated in NEURON from a morphologically and biophysically detailed in vivo-labeled neocortical L5 PC (*Hallermann et al., 2012*). The NEURON model included synapses placed at 200 synaptic locations along the dendritic tree. Although this number of synaptic sites is significantly lower compared to what has been established in biological neurons (*Megías et al., 2001*), this amount of discretization has proven to yield low errors compared to nondiscretized synaptic placements, with fast simulation runtimes and negligible memory consumption (*Figure 5—figure supplement 1*). It is noted that each synaptic location can be contacted by multiple presynaptic cells; therefore, the number of the synaptic locations does not constrain the connectivity. As the computational resource requirements for modeling such complex cells are much higher than in single-compartmental neurons, all NEURON models, data preprocessing, and ANN fitting and query were carried out on single graphical processing units

(GPUs) and tensor processing units (TPUs) ('Methods,' *Figure 5—figure supplement 2*). We found that the trained CNN-LSTM performed in near-perfect accordance with the NEURON simulation (*Figure 5A*, Pearson's $r$ = 0.999, n = 45,000 ms). The continuous self-reliant prediction yielded lower yet adequate AP fidelity (*Figure 5G*, 68.28 ± 18.97% and 66.52 ± 25.37% precision and recall, 0.439 ± 4.181 ms temporal shift for true-positive spikes compared to ground truth, n = 205) compared to the point neuron, and the accuracy of subthreshold membrane potential fluctuations remained high (Pearson's $r$ = 0.83, n = 37).

We previously demonstrated that CNN-LSTMs could accurately predict various neuronal mechanisms beyond somatic voltage fluctuations in single-compartmental cells (*Figure 3*). To investigate whether this architecture is sufficient to describe complex features of neuronal behavior in morphologically and biophysically realistic neurons as well, we tasked the ANN with simultaneously predicting membrane potentials from the soma and two dendritic locations (one apical and one basal) together with calcium current dynamics in the same locations (*Figure 5—figure supplement 3*). We found that CNN-LSTMs can accurately describe the selected aspects of neuronal activity, further demonstrating the versatility of this ANN architecture.

Establishing a proper multicompartmental representation of a neural system by relying solely on the somatic membrane potential is a nontrivial task due to complex signal processing mechanisms taking place in distal subcellular compartments (*Schiller et al., 1997*; *Häusser and Mel, 2003*; *Jarsky et al., 2005*; *Harnett et al., 2015*; *Takahashi et al., 2016*). This is especially true for signals arising from more distal synapses (*Sjöström and Häusser, 2006*; *Larkum et al., 2009*; *Takahashi and Magee, 2009*). To examine whether the CNN-LSTM considered distal inputs or neglected these in favor of more robust proximal inputs, we inspected the weights of the first layer of the neural network architecture (*Figure 5B*). This convolutional layer consists of 512 filters, which directly processes the input matrix (64 ms of 201 input vectors corresponding to the somatic membrane potential and vectorized timing information of 200 synapses). Despite the random initialization of these filters from a uniform distribution (*He et al., 2015*), only a small fraction of optimized filter weights were selected for robust information representation (13.83% of all weights were larger than 0.85), while the majority of them were closer to zero (*Figure 5C*), suggesting relevant feature selection. In order to demonstrate that this feature selection was not biased against distal inputs, the 512 convolutional filters were ranked by their selectivity for distinct synapses. We found that each synaptic input was assigned an independent selectivity filter (*Figure 5D*). Next, we compared the mean weights of each synapse with the somatic amplitude of the elicited voltage response as a proxy for input distance from the soma (*Figure 5E*). This comparison revealed a flat linear correspondence (Pearson's $r$ = 0.06), which combined with the filter specificity (*Figure 5D*) confirmed that distal and proximal synaptic inputs carry equally relevant information for the CNN-LSTM.

When comparing the weights of excitatory and inhibitory inputs, we found that even at the first layer the CNN-LSTM could determine that these inputs have opposing effects on subsequent membrane potential ($5.91 * 10^{-6}$, $2.66 * 10^{-5}$, $-6.22 * 10^{-6}$, and $-1.34 * 10^{-5}$ mean weights for apical excitatory, basal excitatory, apical inhibitory, and basal inhibitory synapses, respectively, n = 51,200, 25,600, 15,360, and 10,240) even though these vectors only contain synaptic conductance information (comparable positive values for both excitatory and inhibitory synapses, *Figure 5F*). Taken together, the feature selectivity and prediction accuracy confirm that the CNN-LSTM architecture is well suited for representing multicompartmental information.

The recent surge in readily available cellular model datasets has significantly reduced the entry barrier for neuronal simulations as researchers no longer need to gather ground truth data individually. Therefore, we aimed to establish a pipeline to constrain ANNs on neuronal models from a publicly available, well-curated database (*Gouwens et al., 2018*) without developer involvement. Using this pipeline, we constrained ANNs on the remaining major cortical PC types; layer 2/3, layer 4, and layer 6 PCs (*Figure 5H*). We found that the resulting ANNs were fit adequately to the NEURON simulations (*Figure 5I*, 94.2 ± 14.2%, 74.5 ± 23.5%, and 67 ± 14.5% variance explained, 86.6 ± 23.1%, 70.1 ± 25.8%, and 63.2 ± 33.2% precision, 90.7 ± 18%, 74.5 ± 25.8%, and 63.5 ± 32.7% recall for layer 2/3, layer 4, and layer 6 PCs, respectively, n = 50), and the fitting procedure was devoid of ambiguities. Together, we developed an ANN architecture appropriate for multicompartmental neuronal simulations of diverse cell types and a user-friendly methodology for their construction.

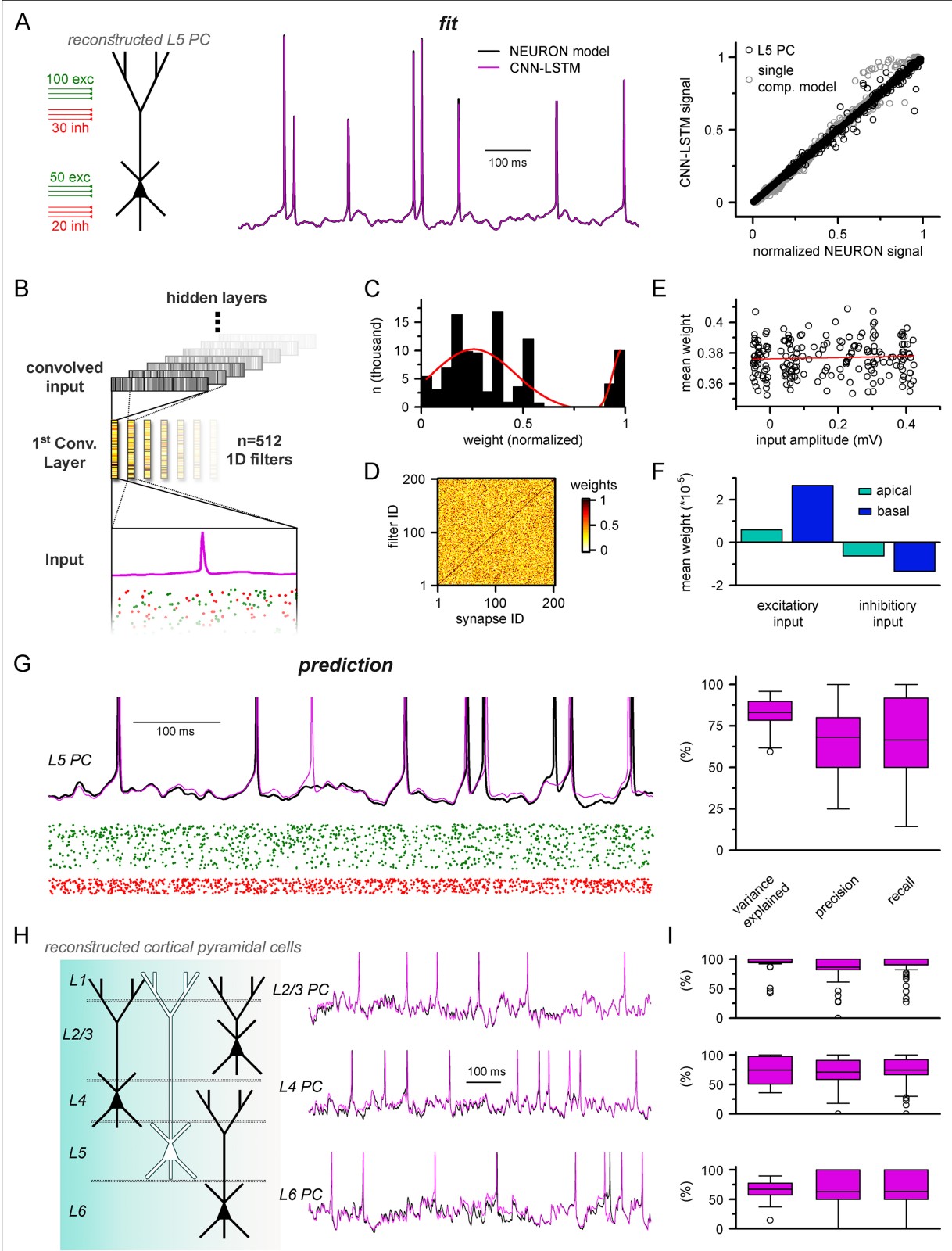

**Figure 5.** Multicompartmental simulation representation by convolutional neural network-long short-term memory (CNN-LSTM).
 (**A**) CNN-LSTM can accurately predict membrane potential of a multicompartmental neuron upon distributed synaptic stimulation. Representative figure depicts the placement of synaptic inputs (150 excitatory inputs: 100 inputs on apical, oblique, and tuft dendrites and 50 inputs on the basal dendrite, randomly distributed; and 50 inhibitory inputs: 30 inputs on apical, oblique, and tuft dendrites and 20 inputs on the basal dendrite,

*Figure 5 continued on next page*

*Figure 5 continued*

randomly distributed) of a reconstructed level 5 (L5) pyramidal cell (PC) (left). Point-by-point forecasting of L5 PC membrane potential by a CNN-LSTM superimposed on biophysically detailed NEURON simulation (left). CNN-LSTM prediction accuracy of multicompartmental membrane dynamics is comparable to single-compartment simulations (right, L5 PC in black, single-compartmental simulation of *Figure 1D* in gray, n = 45,000 and 50,000, respectively). (**B**) Convolutional filter information was gathered from the first convolutional layer (middle, color scale depicts the different weights of the filter), which directly processes the input (membrane potential in magenta, excitatory and inhibitory synapse onsets in green and red, respectively), providing convolved inputs to upper layers (gray bars, showing the transformed 1D outputs). (**C**) Distribution of filter weights from 512 convolutional units (n = 102,400) with double Gaussian fit (red). (**D**) Filter weight is independent of the somatic amplitude of the input (circles are averages from 512 filters, n = 200, linear fit in red). (**E**) Each synapse has a dedicated convolutional unit, shown by plotting the filter weights of the 200 most specific units against 200 synapses. Notice the dark diagonal illustrating high filter weights. (**F**) Excitatory and inhibitory synapse information is convolved by filters with opposing weights (n = 51,200, 25,600, 15,360, and 10,240 for apical excitatory, basal excitatory, apical inhibitory, and basal inhibitory synapses, respectively). (**G**) Representative continuous prediction of L5 PC membrane dynamics by CNN-LSTM (magenta) compared to NEURON simulation (black) upon synaptic stimulation (left, excitatory input in green, inhibitory input in red). Spike timing is measured on subthreshold traces (right, n = 50 for variance explained, precision and recall). (**H**) Artificial neural networks (ANNs) constrained on cortical layer 2/3 (top), layer 4 (middle), and layer 6 (bottom) PCs selected from the Allen Institute model database.

The online version of this article includes the following figure supplement(s) for figure 5:

**Figure supplement 1.** Increased spatial discretization causes reduced errors at the cost of computational overhead.

**Figure supplement 2.** Artificial neural network (ANN) fitting workflow.

**Figure supplement 3.** Convolutional neural network-long short-term memory (CNN-LSTM) predictions of dendritic voltage and current fluctuations of a level 2/3 (L2/3) pyramidal cell (PC).

## Current injection-induced firing responses

The neuronal firing pattern upon direct current injection is one of the most prevalent means of establishing neuronal class and describing the cell's potential in vivo behavior (*Ascoli et al., 2008*). Therefore, these recordings often serve as ground truth data during single-neuronal model constraining (*Izhikevich, 2003*; *Naud et al., 2008*; *Druckmann et al., 2011*; *Teeter et al., 2018*; *Gouwens et al., 2020*). Firing patterns are modulated by several ionic mechanisms in concert, several of which operate on much longer timescales than what the dimensions of our ANN input matrices allow us to observe. However, even complex firing patterns can be approximated by much simpler, biologically plausible, and computationally efficient single-cell models (*Destexhe, 1997*; *Izhikevich, 2003*; *Brette and Gerstner, 2005*; *Sacerdote and Giraudo, 2013*). Therefore, we created a custom ANN layer that can be inserted on top of CNN-LSTMs (for either single- and multicompartmental models) with its internal logic hard-coded based on the governing equations of the eloquent simple spiking model (*Figure 6A*) described by *Izhikevich, 2003*. In addition to the original variables of this model, we set the 'time step' parameter as a variable to account for differences in membrane time constant across cell types.

The custom ANN layer (*Figure 6A*) could reproduce a wide range of naturally occurring firing patterns (*Figure 6B*). In contrast to the millions of free parameters in CNN-LSTMs, this custom layer has only five trainable parameters and thus can be constrained using conventional optimization algorithms (*Singer and Nelder, 2009*). We created a single-compartment NEURON model, equipped with Hodgkin–Huxley conductances based on a fast-spiking phenotype (*Figure 6C*) to generate a ground truth dataset of firing activity and subthreshold membrane potential fluctuations. We found that the custom ANN layer could reliably capture the input–output characteristics of the NEURON model (Pearson's r: 0.982). We next fitted the ANN layer on randomly distributed synaptic inputs (*Figure 6D*). The custom ANN layer produced voltage responses in good agreement with the NEURON model (*Figure 6E and F*, Pearson's r: 0.999, 96.9 ± 0.4% variance explained, n = 17). Together, this custom ANN layer approach imbues CNN-LSTMs with the ability to reproduce firing responses faithfully and also provides added flexibility allowing for the instantaneous alteration of firing behavior while preserving synaptic representations.

Generating diverse custom top layers operating on the output of CNN-LSTMs (*Figure 6—figure supplement 1A*) also creates opportunities to predict convoluted signals used to report neuronal activity in vivo, such as fluorescently reported calcium and voltage signals. To illustrate this possibility, we created custom ANN layers fitted to the dynamics of the GCamp6f fluorescent calcium indicator (*Chen et al., 2013*) and a recently developed fluorescent voltage indicator (*Villette et al., 2019*). Although these indicators severely distorted the underlying neuronal signals (i.e., membrane potential), we found that a custom recurrent encoder can accurately predict these characteristic waveforms

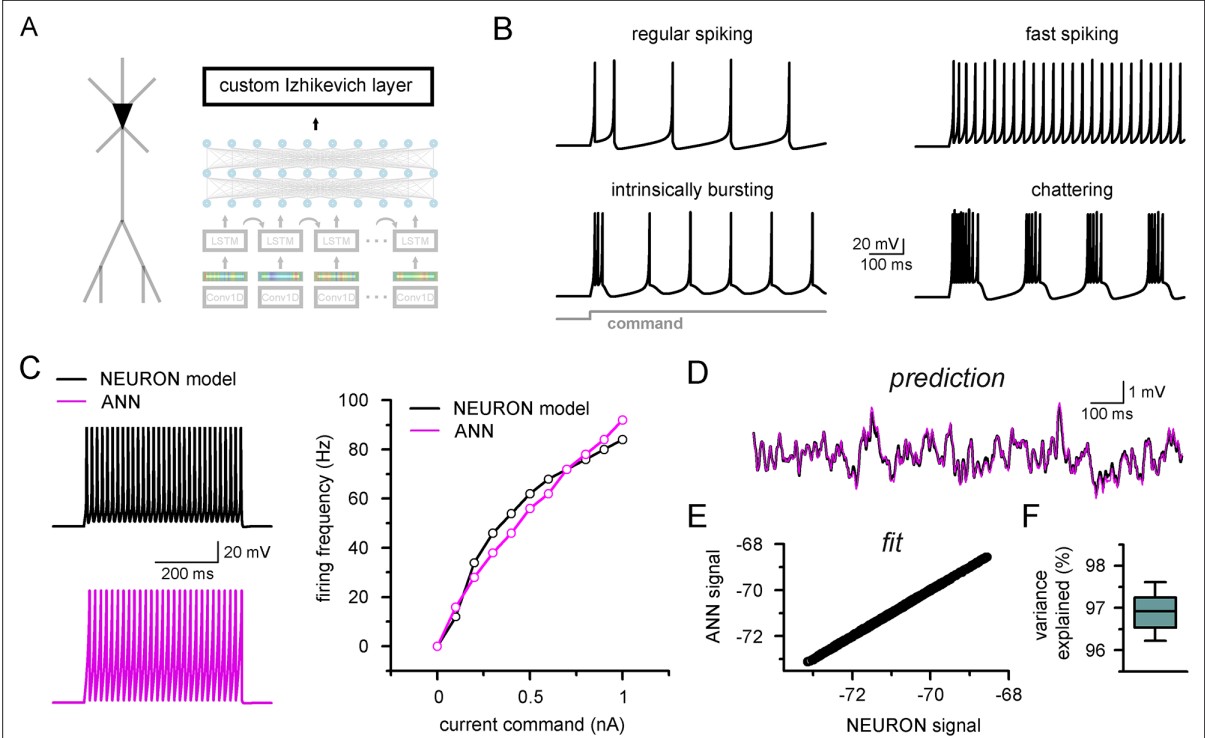

**Figure 6.** Firing pattern representation with custom artificial neural network (ANN) layer.
(**A**) Representative figure depicting the custom ANN layer (termed custom Izhikevich layer) placed on the output of the fully connected layers of the convolutional neural network-long short-term memory (CNN-LSTM). This layer represents the final signal integration step, analogous to the soma of biological neurons. (**B**) Four firing patterns with different activity dynamics, produced by the custom ANN layer. (**C**) Firing pattern of a NEURON model (black, top) and the constrained ANN counterpart (magenta, bottom). The ANN model accurately reproduced the input–output relationship of the NEURON model. (**D**) Continuous subthreshold membrane potential fluctuations of the NEURON model (black trace) and faithfully captured by the custom ANN layer (magenta trace). (**E**) Relationship of membrane potential values predicted step-by-step by the ANN layer compared to the ground truth NEURON model. (**F**) The custom ANN layer continuous predictions explain the majority of the variance occurring in voltage signals produces by the NEURON simulation.

The online version of this article includes the following figure supplement(s) for figure 6:

**Figure supplement 1.** Custom ANN layers for encoding and decoding popular indicators neuronal activity.

---

(*Figure 6—figure supplement 1*), and importantly, stand-alone use of these layers can deconvolve even severely distorted ground truth signals.

## Ultra-rapid simulation of multiple cells using CNN-LSTM

One of the main benefits of this machine learning approach as a substitute for traditional modeling environments is the potential for markedly reduced simulation runtimes. Simulation environments such as NEURON rely on compartment-specific mathematical abstractions of active and passive biophysical mechanisms (*Hines and Carnevale, 1997*), which results in high computational load in increasingly complex circuit models. In the case of small-sized (*Nikolic, 2006*; *Migliore and Shepherd, 2008*; *Cutsuridis and Wennekers, 2009*; *Chadderdon et al., 2014*; *Hay and Segev, 2015*) and mid-sized networks (*Markram et al., 2015*; *Bezaire et al., 2016*; *Shimoura et al., 2018*; *Billeh et al., 2020*) this hinders the possibility of running these models on nonspecialized computational resources. Although several attempts have been made to reduce the demanding computational load of neuronal simulations (*Bush and Sejnowski, 1993*; *Destexhe and Sejnowski, 2001*; *Hendrickson et al., 2011*; *Marasco et al., 2012*; *Rössert, 2016*; *Amsalem et al., 2020*; *Wybo et al., 2021*), the most commonly used approach is parallelization, both at the level of single cells (*Hines et al., 2008*) and network models (*Hines and Carnevale, 2008*; *Lytton et al., 2016*). However, ANNs offer a unique solution to this problem. Contrary to traditional modeling environments, graph-based ANNs are designed

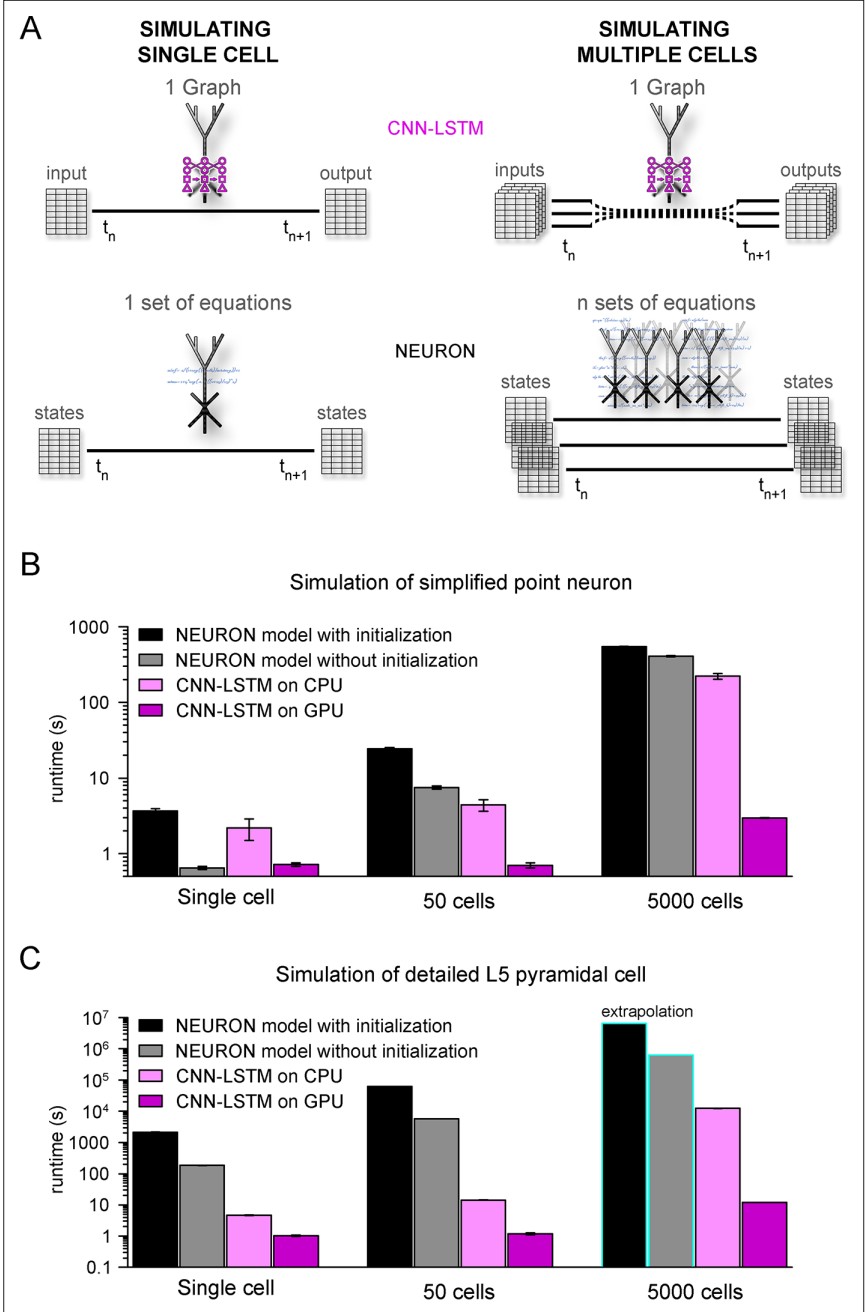

**Figure 7.** Orders of magnitude faster simulation times with convolutional neural network-long short-term memory (CNN-LSTM).
(**A**) An illustration demonstrating that CNN-LSTMs (top, magenta) handle both single-cell (left) and multiple-cell (right) simulations with a single graph, while the set of equations to solve increases linearly for NEURON simulations (bottom, black). (**B**) 100 ms simulation runtimes of 1-, 50-, and 5000-point neurons on four different resources. Bar graphs represent the average of five simulations. (**C**) Same as in panel (B), but for level 5 (L5) pyramidal cell(PC) simulations. Teal borders represent extrapolated datapoints.

explicitly for parallel information processing. This means that ANN simulation runtimes on hardware that enables parallel computing, such as modern GPUs, do not increase linearly after additional cells are integrated into the simulated circuit (*Figure 7A*), resulting in better scaling for large networks where an immense number of similar cells are simulated.

To verify the efficiency of our CNN-LSTM, we compared single cells and small- to mid-sized simulation runtimes against NEURON models used in *Figures 1 and 5*. NEURON simulations were performed

on a single CPU as this is the preferred and most widely used method (but see ; *Ben-Shalom et al., 2022*), while neural nets were run on both CPU and GPU because these calculations are optimized for GPUs. Although GPUs are inherently faster in numerical calculations, NEURON simulations are currently not suitable for this resource; therefore, simulation runtimes were compared using CPUs as well. NEURON simulations were repeated with custom initialization, during which simulations were pre-run to allow time-dependent processes, such as conductance inactivation, to reach steady-state values. Simulation of multiple cells was carried out without the implementation of synaptic connections to establish baseline runtimes, without additional runtime impeding factors. For point neurons, single-cell simulations ran significantly faster in NEURON than their CNN-LSTM counterparts when the optional initialization step was omitted (*Figure 7B*, 3.68 ± 0.24 s, 0.65 ± 0.03 s, 2.19 ± 0.69 ms, and 0.72 ± 0.04 s, 100 ms cellular activity by NEURON with initialization, NEURON without initialization, CNN-LSTM on CPU, and CNN-LSTM on GPU, respectively, n = 5). However, when increasing the number of cells, the predicted optimal scaling of CNN-LSTM models resulted in faster runtimes compared to NEURON models (e.g., for 50 cells, 24.23 ± 1.12 s, 7.45 ± 0.37 s, 4.42 ± 0.77 s, and 0.71 ± 0.05 s for a 100 ms simulation by NEURON with initialization, NEURON without initialization, CNN-LSTM on CPU, and CNN-LSTM on GPU, respectively, n = 5). These results show that while in NEURON the runtimes increased by approximately 6.6 times, CNN-LSTM runtimes on a GPU did not increase.

To demonstrate the practicality of ANNs for typical large-scale network simulations, we repeated these experiments with 5000 cells (representing the number of cells in a large-scale network belonging to the same cell type; *Billeh et al., 2020*). In these conditions, the NEURON simulation was ~148 times slower than a single-cell simulation. Notably, this large-scale CNN-LSTM simulation was only four times slower than that of a single cell (*Figure 7B*, 546.85 ± 4.61 ms, 407.2 ± 9 ms, 222.15458 ± 19.02 ms, and 2.97 ± 0.02ms for simulating 100 ms activity by NEURON with initialization, NEURON without initialization, CNN-LSTM on CPU, and CNN-LSTM on GPU, respectively, n = 5).

We next compared runtime disparities for NEURON and CNN-LSTM simulations of detailed biophysical models (*Figure 7C*). We found that the single-cell simulation of the L5 PC model ran significantly slower than the CNN-LSTM abstraction ($2.08 * 10^3 ± 84.66$ s, 185.5 ± 3.7 s, 4.73 ± 0.13 s, and 1.02 ± 0.05 s for simulating 100 ms activity by NEURON with initialization, NEURON without initialization, CNN-LSTM on CPU, and CNN-LSTM on GPU, respectively, n = 5). This runtime disparity was markedly amplified in simulations with multiple cells (50 cells: $6.3 * 10^4$ s, $5.8 * 10^3$ s, 14.3 ± 0.24 s, and 1.19 ± 0.08 s, 5000 cells: $6.53 * 10^6$ s, $6.28 * 10^5$ s, 901.15 s, and 11.99 s for simulating 100 ms activity by NEURON with initialization, NEURON without initialization, CNN-LSTM on CPU, and CNN-LSTM on GPU respectively, n = 5), resulting in a four to five orders of magnitude faster runtime (depending on initialization) for the CNN-LSTM in case of mid-sized simulations. These results demonstrate that our machine learning approach yields far superior runtimes compared to traditional simulating environments. Furthermore, this acceleration is comparable to that afforded by increased parallel CPU cores used for several network simulations (*Markram et al., 2015*; *Bezaire et al., 2016*; *Billeh et al., 2020*), introducing the possibility of running large or full-scale network simulations on what are now widely available computational resources.

## Efficient parameter space mapping using ANNs

Due to slow simulation runtimes, network simulations are typically carried out only a few times (but see *Barros-Zulaica et al., 2019*), hindering crucial network construction steps, such as parameter space optimization. Therefore, we sought to investigate whether our ANN approach was suitable for exploring parameter space in a pathophysiological system characterized by multidimensional circuit alterations, such as Rett syndrome. Rett syndrome is a neurodevelopmental disorder caused by loss-of-function mutations in the X-linked methyl-CpG binding protein (MeCP2) (*Chahrour and Zoghbi, 2007*). Rett syndrome occurs in ~1:10,000 births worldwide, resulting in intellectual disability, dysmorphisms, declining cortical and motor function, stereotypies, and frequent myoclonic seizures, mostly in girls (*Belichenko et al., 1994*; *Armstrong, 1997*; *Steffenburg et al., 2001*; *Armstrong, 2002*; *Kishi and Macklis, 2004*; *Fukuda et al., 2005*; *Belichenko et al., 2009*). Although the underlying cellular and network mechanisms are largely unknown, changes in synaptic transmission (*Dani et al., 2005*; *Medrihan et al., 2008*; *Zhang et al., 2010*), morphological alterations in neurons (*Akbarian et al., 2001*; *Kishi and Macklis, 2004*), and altered network connectivity (*Dani and Nelson, 2009*) have been reported in Rett syndrome.

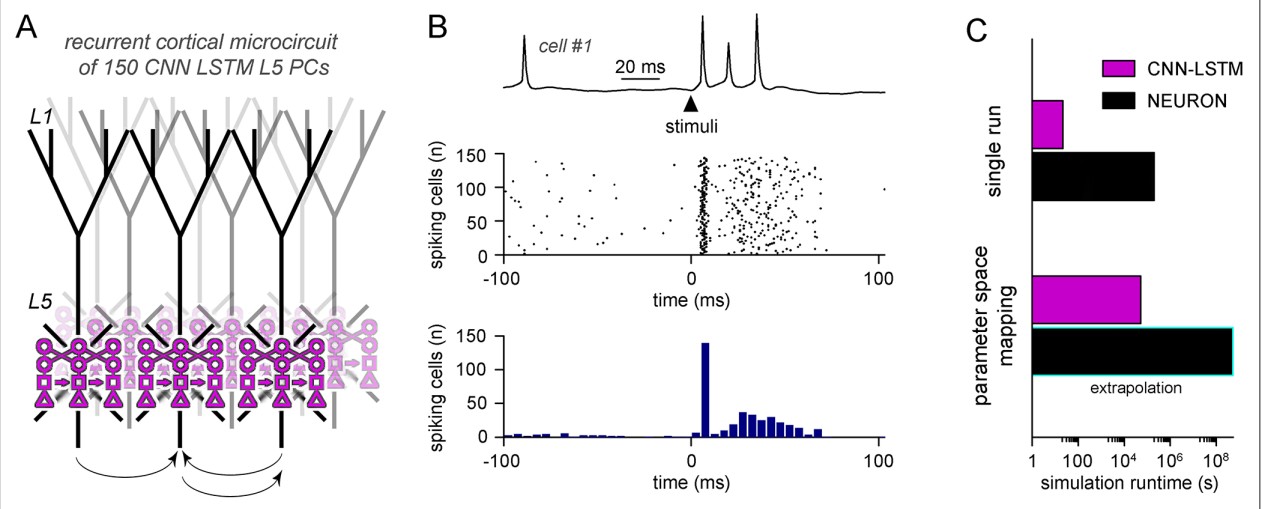

**Figure 8.** Efficient parameter-space mapping with convolutional neural network-long short-term memory (CNN-LSTM) reveals a joint effect of recurrent connectivity and E/I balance on network stability and efficacy in Rett syndrome.
 (**A**) 150 CNN-LSTM models of level 5 (L5) pyramidal cells (PCs) were simulated in a recurrent microcircuit. (**B**) The experimental setup consisted of a stable baseline condition for 100 ms, a thalamocortical input at t = 100 ms, and network response, monitored for 150 ms. Example trace from the first simulated CNN-LSTM L5 PC on top, raster plot of 150 L5 PCs in the middle, number of firing cells with 5 ms binning for the same raster plot in the bottom. Time is aligned to the stimulus onset (t = 0, black arrowhead). (**C**) Simulation runtime for single simulation (left, network of 150 cells simulated for 250 ms) and parameter space mapping (right, 150 cells simulated for 250 ms, 2500 times, for generating **B**). Teal border represents data extrapolation.

We aimed to investigate the contribution of the distinct alterations on cortical circuit activity in Rett syndrome using a recurrent L5 PC network (*Hay and Segev, 2015*) composed entirely of CNN-LSTM-L5-PCs (*Figure 8A*). Simulations were run uninterrupted for 100 ms when a brief (1 ms) perisomatic excitation was delivered to mimic thalamocortical input onto thick tufted PCs (*de Kock et al., 2007*; *Meyer et al., 2010*; *Constantinople and Bruno, 2013*). In control conditions, cells fired well-timed APs rapidly after the initial stimuli followed by an extended AP firing as a consequence of the circuit recurrent connectivity (*Figure 8B*; *Lien and Scanziani, 2013*; *Sun et al., 2013*). First, we compared the runtime of the simulated L5 microcircuit of CNN-LSTM models and the run time of 150 unconnected L5 PCs in NEURON. We found that for a single simulation, CNN-LSTM models were more than 9300 times faster compared to NEURON models (*Figure 8C*, 21.153 ± 0.26 s vs. 54.69 hr for CNN-LSTM and NEURON models, respectively).

## Rett cortical network alterations counteract circuit hyperexcitability

Cortical networks endowed with frequent recurrent connections between excitatory principal cells are prone to exhibit oscillatory behavior, which is often the mechanistic basis of pathophysiological network activities (*McCormick and Contreras, 2001*; *Figure 9A*). We quantified oscillatory activity (*D'Cruz et al., 2010*; *McLeod et al., 2013*; *Roche et al., 2019*) and the immediate response to thalamocortical stimuli independently (*Figure 8C*). By systematically changing excitatory quantal size (*Dani et al., 2005*) and the ratio of recurrent L5 PC innervation to mimic reduced recurrent connectivity and synaptic drive in Rett syndrome, we found that both alterations had considerable influence over network instability (*Figure 9B*, left panel; excitatory drive: 17.85 ± 61.61 vs. 388.92 ± 170.03 pre-stimulus APs for excitatory drive scaled by 0.75 and 1.25, respectively, n = 100 each, p<0.001; recurrent connectivity: 321.96 ± 200.42 vs. 157.66 ± 192.5 pre-stimulus APs for 10 and 5.2% recurrent connectivity, similar to reported values for adult wild-type and *Mecp2*-null mutant mice [*Dani and Nelson, 2009*], n = 50 each, p<0.001) and response to stimuli (excitatory drive: 147.58 ± 17.2 vs. 119.23 ± 18.1 APs upon stimulus for excitatory drive scaled by 0.75 and 1.25, respectively, n = 100 each, p=2.3 * $10^{-22}$, t(198) = 11.03, *two-sample t-test*; recurrent connectivity: 134.76 ± 21.37 vs. 112.74 ± 34.99 APs upon stimulus for 10 and 5.2% recurrent connectivity, n = 50 each, p=2.54 * $10^{-4}$, t(98) = 3.8, *two-sample t-test*). Contrary to disruption of the excitatory drive, when inhibitory quantal size (*Chao et al., 2010*) was altered, we found that inhibition had a negligible effect on

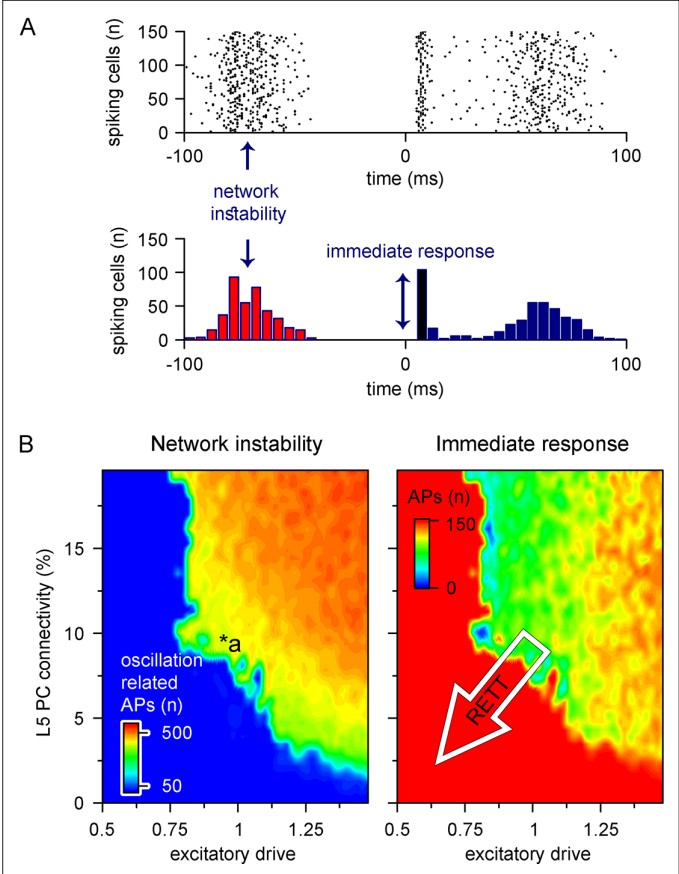

**Figure 9.** Recurrent connectivity and excitatory drive jointly define network stability in a reduced level 5 (L5) cortical network.
 (**A**) Two independent parameters were quantified: network instability (number of cells firing before the stimulus) and immediate response (number of cells firing within 10 ms of the stimulus onset). The example simulation depicts highly unstable network conditions. (**B**) Network instability (left) and immediate response (right) as a function of altered L5 pyramidal cell (PC) connectivity and excitatory drive. *a indicates network parameters used for generating panel (**A**). The white arrow in the right panel denotes circuit alterations observed in Rett syndrome. Namely, 5% recurrent connectivity between L5 PCs instead of 10% in control conditions and reduced excitatory drive.

The online version of this article includes the following figure supplement(s) for figure 9:

**Figure supplement 1.** Microcircuit stability and efficacy is robust to changes in inhibitory drive.

network instability, as connectivity below 9% never resulted in oscillatory activity (*Figure 9—figure supplement 1*; inhibition corresponds to random inhibitory drive, as the network did not contain ANNs representing feed-forward inhibitory cells). Interestingly, we found no measurable relationship between the inhibitory quantal size and the network response to thalamocortical stimuli either. These results suggest that lowered recurrent connectivity reduces network instability. Specifically, recurrent connectivity observed in young *Mecp2*-null mice (7.8%; *Dani and Nelson, 2009*) yielded more stable microcircuits (54% of networks were stable, n = 100) than wild-type conditions (34% of networks were stable, n = 50). Recurrent connection probability of older animals (5.3%) further stabilized this network (64% of networks were stable). Taken together, our model suggests that reduced recurrent connectivity between L5 PCs is not causal to seizure generation and abnormal network activity (*Steffenburg et al., 2001*; *Roche et al., 2019*), which are crucial symptoms of Rett syndrome at a young age, but instead normal PC activation is disrupted. This may correspond to the early stages of Rett syndrome where cortical dysfunction emerges before the appearance of seizures (*Chahrour and Zoghbi, 2007*).

Using the ANN approach, we successfully implemented multidimensional parameter space mapping in a cortical circuit exhibiting pathophysiological changes and could identify the isolated

outcome of distinct circuit alterations. Furthermore, our accelerated multicompartmental neural circuit model demonstrated that parameter space mapping is not only attainable by CNN-LSTM models on commercially available computational resources, but it is almost fourfold faster than completing a single NEURON simulation.

## Discussion

In this study, we present an ANN architecture (CNN-LSTM) capable of accurately capturing neuronal membrane dynamics. Most of the investigated ANN architectures predicted subthreshold voltage fluctuations of point neurons; however, only the CNN-LSTM was able to generate APs. This model could generalize well to novel input and also predict various other features of neuronal cells, such as voltage-dependent ionic current dynamics. Furthermore, the CNN-LSTM accounted for the majority of the variance of subthreshold voltage fluctuations of biophysically realistic L5 PC models with excitatory and inhibitory synapses distributed along the entirety of the dendritic tree. The timing of the predicted APs closely matched the ground truth data. Importantly, we found that the CNN-LSTM has superior scaling for large network simulations. Specifically, in the case of mid-sized biophysically detailed networks (50 cells), ANNs were more than three orders of magnitude faster, while for large-scale networks (5000 cells) ANNs are predicted to be five orders of magnitude faster than traditional modeling systems. These accelerated simulation runtimes allowed us to quickly investigate an L5 PC network in distinct conditions, for example, to uncover network effects of altered connectivity and synaptic signaling observed in Rett syndrome. In our Rett cortical circuit model, recurrent connectivity and excitatory drive jointly shape network stability and responses to sensory stimuli, showing the power of this approach in generating testable hypotheses for further empirical work. Together, the described model architecture provides a suitable alternative to traditional modeling environments with superior simulation speed for biophysically detailed cellular network simulations.

### Advantages and limitations of the CNN-LSTM architecture

As our familiarity with neuronal circuits grows, so does the complexity of models tasked with describing their activity. Consequently, supercomputers are a regular occurrence in research articles that describe large-scale network dynamics built upon morphologically and biophysically detailed neuronal models (*Markram et al., 2015*; *Bezaire et al., 2016*; *Billeh et al., 2020*). Here, we developed an alternative to these traditional models, which can accurately represent the full dynamic range of neuronal membrane voltages in multicompartmental cells, but with substantially accelerated simulation runtimes.

ANNs are ideal substitutes for traditional model systems for several reasons. First, ANNs do not require hard-coding of the governing rules for neuronal signal processing. Upon creation, ANNs serve as a blank canvas that can derive the main principles of input–output processing and neglect otherwise unimpactful processes (*Benitez et al., 1997*; *Dayhoff and DeLeo, 2001*; *Castelvecchi, 2016*). The degree of simplification depends only on the ANN itself, not the developer, thereby reducing human errors. However, architecture construction and training dataset availability represent limiting steps in ANN development (*Alwosheel et al., 2018*). Fortunately, the latter issue is void as virtually infinite neuronal activity training datasets are now available for deep learning. Conversely, as we have demonstrated, the former concern can significantly impede ANN construction. Although we have shown that markedly divergent ANN architectures can accurately depict subthreshold signal processing, we found only one suitable for both subthreshold active membrane potential prediction. The presented architecture is unlikely to be the only suitable ANN model for neural simulations as machine learning is a rapidly progressing field that frequently generates highly divergent ANN constructs (*da Silva et al., 2017*). The importance of the network architecture is further emphasized by our findings demonstrating that ANNs with comparable or even greater numbers of freely adjustable parameters could not handle suprathreshold information.

The prevailing CNN-LSTM architecture was proven suitable for depicting membrane potential and ionic current dynamics of both simplified and biophysically detailed neuronal models and generalized well for previously unobserved simulation conditions. These results indicate that ANNs are ideal substitutes for traditional model systems for representing various features of neuronal information processing with significantly accelerated simulations. Future architecture alterations should focus on

the continued improvement of AP timing and prediction, as well as the integration of additional dendritic and axonal properties.

A recent publication presented an excellent implementation of an ANN architecture for predicting neuronal membrane potentials (*Beniaguev et al., 2021*) of complex cortical neurons. The featured architecture was composed of nested convolutional layers, and membrane potential dynamics was represented with a combination of two output vectors (subthreshold membrane potential and a binarized vector for AP timing). Building on this idea, we aimed to design an architecture that could (1) produce sequential output with smaller temporal increments, (2) generalize to previously unobserved temporal patterns and discrepant synaptic weights as well, and lastly, (3) produce APs with plausible waveforms in addition to subthreshold signals. Fulfillment of these three criteria is imperative for modeling these cells in a network environment. Our ANN architecture fulfills these requirements, thus representing the first ANN implementation that can serve as a viable alternate for biophysically and morphologically realistic neurons in a network model environment.

The CNN-LSTM architecture has several advantages over traditional modeling environments beyond the runtime acceleration. For example, connectivity has no influence over simulation speed as connection implementation is a basic matrix transformation carried out on the entire population simultaneously. However, this approach is not without limitations. First, although ANN training can be carried out on affordable and widely available resources, training times can last up to 24 hr to achieve accurate fits ('Methods'). Furthermore, judicious restrictions are needed in the amount of synaptic contact sites, to preserve realistic responses and at the same time mitigate computational requirements, as the number of contact sites directly correlates with simulation runtimes and memory consumption. Additionally, the 1 ms temporal discretization hinders the implementation of certain biological phenomena that operate on much faster timescales, such as gap junctions. Depending on the degree of justifiable simplification, several other modeling environments exist, which are faster and computationally less demanding than our ANN approach. These environments mostly rely on simplified point neurons, such as the Izhikevich formulation (*Figure 6*), often developed specifically to leverage accelerated GPU computations (*Ros et al., 2006*; *Fidjeland et al., 2009*; *Nageswaran et al., 2009*; *Mutch, 2010*; *Thibeault, 2011*; *Nowotny et al., 2014*; *Vitay et al., 2015*; *Yavuz et al., 2016*; *Knight et al., 2021*). Therefore, depending on the required biophysical resolution and the available computational resources, the ANN approach presented here has an advantage over other environments in certain situations, while traditional modeling environments such as NEURON and GPU accelerated network simulators have a distinct edge in other use cases.

## Simulation runtime acceleration

Accelerated simulation runtimes are particularly advantageous for large-scale biological network simulations, which have seen an unprecedented surge in recent years. These network simulations not only provide support for experimentally gathered information but also as testing benchmarks in the future for several network-related queries such as pharmaceutical target testing and for systemic interrogation of cellular-level abnormalities in pathophysiological conditions (*Gambazzi et al., 2010*; *Kerr et al., 2013*; *Neymotin et al., 2016a*, *Sanjay, 2017*; *Domanski et al., 2019*; *Zhang and Santaniello, 2019*; *Liou et al., 2020*). However, widespread adaptation of large-scale network simulations is hindered by the computational demand of these models that can only be satisfied by the employment of supercomputer clusters (*Markram et al., 2015*; *Bezaire et al., 2016*; *Billeh et al., 2020*). Because these resources are expensive, they do not constitute a justifiable option for general practice. Importantly, we have shown that ANNs can provide a suitable alternative to traditional modeling systems, and that their simulation runtimes are also superior due to the structure of the machine learning platform (i.e., Tensorflow).

Traditional model systems linearly increase the number of equations to be solved for parallelly simulated cells, while ANNs can handle cells belonging to the same cell type on the same ANN graph (*Dillon, 2017*). In our network models (150 cells; *Figure 8*), NEURON simulations yield 150 times more linear equations for every time step, while ANNs used the same graph for all simulated cells. This property of ANNs in particular suits biological networks consisting of many cells. For example, the Allen Institute reported a computational model of the mouse V1 cortical area (*Billeh et al., 2020*), consisting of 114 models corresponding to 17 different cell types (with the number of cells corresponding to these cell types ranging from hundreds to more than 10,000), which means

that simulation of a complete cortical area is feasible using only 114 ANNs. We have demonstrated that even for small networks consisting of only 150 cells of the same type ANNs are more than four orders of magnitude faster compared to model environments used in the aforementioned V1 simulations. As large-scale network simulations are typically run using several thousand CPU cores in parallel, the predicted runtime acceleration suggests that network simulations relying on ANNs could negate the need for supercomputers. Instead, ANN-equivalent models could be run on commercially available computational resources such as personal computers with reasonable time frames.

Another advantage of our approach is the utilization of GPU processing, which provides a substantially larger number of processing cores (*Asano et al., 2009*; *Memon et al., 2017*). The runtime differences are observable by comparing CNN-LSTM simulations on CPU and GPU (*Figure 7B and C*), which yields more than an order of magnitude faster simulations on GPU in the case of small-size networks (50 cells) and approximately two orders of magnitude difference for mid-sized networks. Our results demonstrate that cortical PC network simulations are at least four orders of magnitude faster than traditional modeling environments, confirming that disparities in the number of cores can only partially account for the observed ANN runtime acceleration. Furthermore, the NEURON simulation environment does not benefit as much from GPU processing as for ANN simulations (*Vooturi et al., 2017*; *Kumbhar et al., 2019*). These results confirm that the drastic runtime acceleration is the direct consequence of the parallelized graph-based ANN approach.

## Efficient mapping of network parameter involvement in complex pathophysiological conditions

To demonstrate the superiority of ANNs in a biologically relevant network simulation, we mapped the effects of variable network parameters observed in Rett syndrome. Rett syndrome is a neurodevelopmental disorder leading to a loss of cognitive and motor functions, impaired social interactions, and seizures in young females due to loss-of-function mutations in the X-linked *MeCP2* gene (*Chahrour and Zoghbi, 2007*). Like many brain diseases, these behavioral alterations are likely due to changes in several different synaptic and circuit parameters. MeCP2-deficient mice exhibit multiple changes in synaptic communication, affecting both excitatory and inhibitory neurotransmission and circuit-level connectivity. Excitatory transmission is bidirectionally modulated by *MeCP2* knockout (*Nelson et al., 2006*; *Chao et al., 2007*) and overexpression (*Na et al., 2012*), and long-term synaptic plasticity is also impaired in MeCP2-deficient mice (*Asaka et al., 2006*; *Guy et al., 2007*). Inhibitory signaling is also altered in several different brain areas (*Dani et al., 2005*; *Medrihan et al., 2008*). Importantly, synaptic transmission is affected not only at the level of quantal parameters but also regarding synaptic connections as MeCP2 directly regulates the number of glutamatergic synapses (*Chao et al., 2007*). This regulation amounts to a 39% reduction of putative excitatory synapses in the hippocampus (*Chao et al., 2007*) and a 50% reduction in recurrent excitatory connections between L5 PCs (*Dani and Nelson, 2009*). Here, we investigated how these diverse underlying mechanisms contribute to overall circuit pathology using our ANN network model approach.

We found that the ability of the network to respond to external stimuli is affected by both alterations in synaptic excitation and changes in the recurrent connectivity of L5 PCs. Our results suggest that disruption of inhibitory transmission is not necessary to elicit network instability in Rett as changes in synaptic excitation and recurrent connectivity alone were sufficient in destabilizing the network. These results are supported by previous findings showing that both constitutive (*Calfa et al., 2011*) and excitatory-cell-targeted (*Zhang et al., 2014*) *MeCP2* mutations lead to network seizure generation as opposed to inhibitory-cell-targeted *MeCP2* mutation, which causes frequent hyperexcitability discharges but never seizures (*Chao et al., 2010*). Furthermore, our results suggest that excitatory synaptic alterations in Rett affect both general network responses and network stability, which may serve as substrates to cognitive dysfunction and seizures, respectively. Taken together, our results reveal how cellular-synaptic mechanisms may relate to symptoms at the behavioral level. Importantly, investigation of the multidimensional parameter space was made possible by the significantly reduced simulation times of our ANN as identical simulations with traditional modeling systems are proposed to be four orders of magnitude slower.

## Methods

### Single-compartmental NEURON simulation

Passive and active membrane responses to synaptic inputs were simulated in NEURON (*Hines and Carnevale, 1997*, version 7.7, available at http://www.neuron.yale.edu/neuron/). Morphology (single compartment with length and diameter of 25 μm) and passive cellular parameters ($R_m$: 1 kΩ/cm²; $C_m$: 1 μF/cm²; $R_i$: 35.4 Ω/cm) were the same for both cases and resting membrane potential was set to –70 mV. Additionally, the built-in mixed sodium, potassium and leak channel (*Jaslove, 1992*, based on the original Hodgkin–Huxley descriptions) was included in the active model ($g_{Na}$: 0.12 pS/μm²; $g_K$: 0.036 pS/μm²; $g_{leak}$: 0.3 nS/μm²). Reversal potentials were set to 50 mV for sodium, –77 mV for potassium, and –54.3 mV for leak conductance. Simulations were run with a custom steady-state initialization procedure (*Carnevale and Hines, 2006*) for 2 s, after which the temporal integration step size was set to 25 μs.

In order to simulate membrane responses to excitatory and inhibitory inputs, the built-in AlphaSynapse class of NEURON was used (excitatory synapse: $\tau$: 2 ms; $g_{pas}$: 2.5 nS; $E_{rev}$: 0 mV; inhibitory synapse: $\tau$: 1 ms; $g_{pas}$: 8 nS; $E_{rev}$: –90 mV). The number of synapses was determined by a pseudo-random uniform number generator (ratio of excitatory to inhibitory synapses: 8:3). Timing of individual synapses was also randomly picked from a uniform distribution. During the 10-s-long simulations, the membrane potential, $I_{Na}$, and $I_K$ currents were recorded along with the input timings and weights and were subsequently saved to text files. Simulations were carried out in three different conditions. First, resting membrane potential was recorded without synaptic activity. Second, passive membrane potential was recorded. Third, active membrane potential responses were recorded with fixed synaptic weights.

The amount of training each ANN received varied widely, based on the complexity of the modeled system. We used model checkpoints to stop the training if the prediction error on the validation dataset did not improve within 20 training epochs. This checkpoint was reached between 12 and 24 hr, training on a single GPU.

### Multicompartmental NEURON simulation

Active multicompartmental simulations were carried out using an in vivo-labeled and fully reconstructed thick tufted cortical L5 PC (*Hallermann et al., 2012*). The biophysical properties were unchanged, and a class representation was created for network simulations. Excitatory and inhibitory synapses were handled similarly to single-compartmental simulations. A total of 100 excitatory ($\tau$: 1 ms; $g_{pas}$: 3.6 nS; $E_{rev}$: 0 mV) and 30 inhibitory synapses ($\tau$: 1 ms; $g_{pas}$: 3 nS; $E_{rev}$: –90 mV) were placed on the apical, oblique, or tuft dendrites, and 50 excitatory and 20 inhibitory synapses were placed on basal dendrites. The placement of the synapses was governed by two uniform pseudo-random number generators, which selected dendritic segments weighed by their respective lengths and the location along the segment (ratio 2:1:1:1 for apical excitatory, apical inhibitory, basal excitatory, and basal inhibitory synapses). Simulations were carried out with varied synaptic weights and a wide range of synapse numbers.

### ANN benchmarking

MTSF models are ideal candidates for modeling neuronal behavior in a stepwise manner as they can be designed to receive information about past synaptic inputs and membrane potentials in order to predict subsequent voltage responses. These ANNs have recently been demonstrated to be superior to other algorithms in handling multivariate temporal data such as audio signals (*Kons and Toledo-Ronen, 2013*), natural language (*Collobert and Weston, 2008*), and various other types of fluctuating time-series datasets (*Zheng et al., 2014*; *Che et al., 2018*; *Zhang et al., 2019*). To validate the overall suitability of different ANN architectures tested in this article for MTSF, we used a weather time-series dataset recorded by the Max Planck Institute for Biogeochemistry. The dataset contains 14 different features, including humidity, temperature, and atmospheric pressure collected every 10 min. The dataset was prepared by François Chollet for his book *Deep Learning with Python* (dataset preparation steps can be found on the Tensorflow website: https://www.tensorflow.org/tutorials/structured_data/time_series). All ANN architectures were implemented using the Keras deep-learning API (https://keras.io/) of the Tensorflow open-source library (version 2.3, *Abadi, 2015*;, https://www.tensorflow.org/), with Python 3.7.

The first architecture we implemented was a simple linear model consisting of three layers without activation functions; a Flatten layer, a Dense (fully connected) layer with 64 units, and a Dense layer with 3 units. The second architecture was a linear model with added nonlinear processing. The model contained three layers identical to the linear model, but the second layer had a sigmoid activation function. The third model was a deep neural net with mixed linear and nonlinear layers. Similar to the first two models, this architecture had a Flatten layer and a Dense layer with 64 units as the first two layers, followed by nine Dense layers (units 128, 256, 512, 1024, 1024, 512, 256, 128, and 64 for the nine Dense layers) with hyperbolic tangent (tanh) activation function and Dropout layers with 0.15 dropout rate. The last layer was the same Dense layer with three units as in case of the linear and nonlinear models. The fourth model was a modified version of the WaveNet architecture introduced in 2016 (*Oord, 2016*), implemented based on a previous publication (*Beniaguev et al., 2021*). The fifth and final architecture was a convolutional LSTM model (*Donahue et al., 2015*) that consists of three distinct functional layer segments. The lowest layers (close to the input layer) were three, one-dimensional convolutional layers (Conv1D) with 128, 100, and 50 units, and causal padding for temporal data processing. The first and third layers had a kernel size of 1, and the second layer had a kernel size of 5. The first two layers had 'rectified linear unit' (relu) activation functions, and the third layer had tanh activation; therefore, the first two layers were initialized by He-uniform variance scaling initializers (*He et al., 2015*), while the third layer was initialized by Glorot-uniform initialization (also known as Xavier uniform initialization) (*Glorot, 2011*). After flattening and repeating the output of this functional unit, a single LSTM layer (*Hochreiter and Schmidhuber, 1997*) handled the arriving input, providing recurrent information processing. This layer had 128 units, tanh activation function, Glorot-uniform initialization, and was tasked to return sequences instead of the last output. The final functional unit was composed of four Dense layers with 100 units, scaled exponential linear unit (selu) activations, and accordingly, LeCun-uniform initializations (*Montavon et al., 2012*). The dropout rate between Dense layers was set to 0.15.

All benchmarked architectures were compiled and fitted with the same protocol. During compiling, the loss function was set to calculate mean squared error and the Adam algorithm (*Kingma and Ba, 2014*) was chosen as the optimizer. The maximum number of epochs was set to 20; however, an early stopping protocol was defined to have a patience of 10, which was reached in all cases.

## Single-compartmental simulation representation with ANNs

As neural nets favor processed data scaled between –1 and 1 or 0 and 1, we normalized the recorded membrane potentials and ionic currents. Due to the 1 Hz recording frequency, AP amplitudes were variable beyond physiologically plausible ranges; therefore, peak amplitudes were standardized. The trainable time-series data was consisting of 64-ms-long input matrices with three or five columns (corresponding to membrane potential, excitatory input, inhibitory input, and optionally $I_{Na}$ and $I_K$ current recordings) and target sequences were vectors with one or three elements (membrane potential and optional ionic currents). Training, testing, and validation datasets were created by splitting time-series samples 80-10–10%.

Benchmarking the five different ANN architectures proved that these models can handle time-series data predicting with similar accuracy; however, in order to obtain the best results, several optimization steps of the hyperparameter space were undertaken. Unless stated otherwise, layer and optimization parameters were unchanged compared to benchmarking procedures. First, linear models were created without a Flatten layer, instead of which a TimeDistributed wrapper was applied to the first Dense layer. The same changes were employed in case of the nonlinear model and the deep neural net. The fourth, convolutional model had 12 Conv1D layers with 128 filters, kernel size of 2, causal padding tanh activation function and dilatation rates constantly increasing by $2^n$. We found that the best optimization algorithm for passive and active membrane potential prediction is the Adam optimizer accelerated with Nesterov momentum (*Dozat, 2015*), with gradient clipping set to 1. Although mean absolute error and mean absolute percentage error were sufficient for passive membrane potential prediction, the active version warranted the usage of mean squared error in order to put emphasis on APs. We found out that the mechanistic inference of the full dynamic range of simulated neurons was a hard task for ANNs; therefore, we sequentially trained these models in a specific order. First, we taught the resting membrane potential by supplying voltage recordings with only a few or no synaptic inputs. This step was also useful to learn the isolated shapes of

certain inputs. Second, we supplied highly active subthreshold membrane traces to the models and finally inputted suprathreshold membrane potential recordings. During the subsequent training steps, previous learning phases were mixed into the new training dataset in order to avoid the catastrophic forgetting of gradient-based neural networks (**Goodfellow, 2015**).

During altered excitation–inhibition ratios, the previously constructed single-compartmental model was used without modifications in layer weights and biases. Firing responses were fitted with different curves, a linear model,

$$y = a + bx$$

which could account for either subtractive or divisive inhibition (**Bhatia et al., 2019**), and a logistic curve,

$$y = \frac{A1 - A2}{1 + e^{(x - x0)/dx}} + A2$$

representing divisive normalization. Although the latter arithmetic operation is often approximated by an exponential curve, we felt the necessity to account for datapoints without spiking.

In experiments aimed at quantifying the effect of biophysical modifications of delayed rectifier potassium conductances, left- and right-shifted models were compared to control conditions point-by-point upon identical synaptic input streams, and the deviation from control conditions was expressed as absolute difference, measured in millivolts.

NMDA point-process model was constructed as a compound model consisting of an AMPA and an NMDA segment, both of which were designed based on NEURON's built-in AlphaSynapse class. The logic of the model was based on a previous publication (**Kim et al., 2013**), where the AMPA model was only dependent on local membrane potential, while the NMDA model had an additional constraining Boltzmann function for gating voltage-dependent activation. The ANN was trained on several datasets having consistently higher randomly distributed synaptic inputs. The training dataset did not contain activity patterns tested in **Figure 4**. The training dataset consisted of an nX4 matrix, where the columns were membrane voltage, AMPA conductance, NMDA conductance, and inhibitory conductance. In the training dataset, AMPA and NMDA synapses were applied independently, and the Boltzmann function of NMDA was omitted. After the model learned the correct representation of NDMA activations, a hand-crafted layer was inserted into the ANN, which recalculated the conductance maximum of NMDA in accordance with the instantaneous membrane potential. Specifically, the function was expressed as

$$gNDMA = \frac{A1 - A2}{1 + e^{(v - x0)/dx}} + A2$$

where *A1* is 1, *A2* is –1, *v* denotes membrane potential, *x0* is set to –63.32 in NEURON and 1.44 in the ANN, while *dx* is 0.013 in NEURON and 0.12 in the ANN.

## CCN-LSTM for multicompartmental simulation representation

Data preprocessing was done as described for single-compartmental representations. Time-series data for CNN-LSTM input was prepared as matrices having 201 rows for membrane potential and 200 synapse vectors, and 64 rows (64-ms-long input). The CNN-LSTM architecture consisted of three Conv1d layers (512, 256, and 128 units), a Flatten layer, a RepeatVector, three LSTM layers (128 units each), and six Dense layers (128, 100, 100, 100, 100, 1 units). Activation functions and initializations were similar to the CNN-LSTM described above, with the exception of the first Dense layer, which included the relu activation function and He-uniform initialization. Additionally, Lasso regularization (**Santosa and Symes, 1986**) was applied to the first Conv1D layer. We found that the best optimizer for our purposes was a variant of the Adam optimizer based on the infinity norm, called Adamax (**Kingma and Ba, 2014**). Due to the non-normal distribution of the predicted membrane potentials, an inherent bias was present in our results, which was scaled by either an additional bias term, or a nonlinear function transformation.

Network construction was based on a previous publication (**Hay and Segev, 2015**). Briefly, 150 L5 PC were simulated in a network with varying unidirectional connectivity, and bidirectional connectivity proportional to the unidirectional connectivity ($P_{bidirectional} = 0.5 * P_{unidirecional}$). Reciprocal connections were 1.5 times stronger than unidirectional connections. In order to implement connectivity,

a connection matrix was created, where presynaptic cells corresponded to the rows, and postsynaptic cells corresponded to the columns of the matrix. If there was a connection between two cells, the appropriate element of the matrix was set to 1, otherwise the matrix contained zeros. Next, cells were initialized with random input matrices. After a prediction was made for the subsequent membrane potential values, every cell was tested for suprathreshold activity. Upon spiking, rows of the connectivity matrix corresponding to the firing cells were selected, and the input matrices of the postsynaptic cells were supplemented with $x_{ij} * g_{conn}$ , where $x_{ij}$ corresponds to the element of the connectivity matrix for presynaptic cell $i$, and postsynaptic cells $j$, and $g_{conn}$ refers to the conductance of the synapses between two connected cells. As this step is carried out upon presynaptic spiking, regardless of whether two cells are connected or not ($x_{ij}$ can be 0 or 1), the degree of connectivity does not influence simulation runtimes.

The delay between presynaptic AP at the soma and the onset of the postsynaptic response was 1 ms measured from the AP peak as the network simulations represent local circuit activity. If the simulated network is made to include spatially circuit components with more variability in their synaptic delays, to account for their spatial segregation, a buffer matrix must be created. The aim of this buffer matrix is to contain synaptic conductance values upon AP detection from the presynaptic cells, without immediately posting it on the input matrices of postsynaptic cells. Each connection consisted of five proximal contact sites. Compared to the original publication, we modified the parameters of the Tsodyks–Markram model (*Tsodyks and Markram, 1997*) used to govern synaptic transmission and plasticity. Based on a recent publication (*Barros-Zulaica et al., 2019*), we set U (fraction on synaptic resources used by a single spike) to 0.38, D (time constant for recovery from depression) to 365.6, and F (time constant for recovery from facilitation) to 25.71. The simulation was run for 250 or 300 ms, which consisted of a pre-stimuli period (to observe the occurrence of structured activity patterns) for 100 ms, and a post-stimuli period (to quantify network amplification). The stimulus itself consisted of a strong excitatory input (can be translated to 50 nS) delivered to a proximal dendritic segment, calibrated to elicit APs from all 150 cells in a 10-ms-long time window. Scaling of inhibitory inputs was carried out by changing inhibitory quantal size of background inputs, while scaling of excitatory drive affected quantal size of recurrent synaptic connections as well.

## Custom top layers

We created custom top layers operating on the output layer of the CNN-LSTM in two different configurations, First, the 'custom Izhikevich layer' was implemented using the 'CustomLayer' class of Tensorflow. The internal variables and governing functions were implemented based on the original description of this model (*Izhikevich, 2003*). Briefly, the layer calculates the values of v and u dimensionless variables (v represents membrane potential, and u represents a membrane recovery variable), based on a, b, c, and d dimensionless parameters (a corresponds to the timescale of u, b sets the sensitivity of u, c describes the after-spike reset value of v, and d sets the after-spike reset value of u). Additionally, we set dt (time step) parameter free as it was necessary for accounting for the membrane time constant. Due to the low number of trainable parameters, this layer can be fitted with conventional fitting algorithms, such as the Nelder–Mead minimalization (*Singer and Nelder, 2009*), available in the 'scipy' package of Python. As the Izhikevich equations require information about the state of both u and v variable, yet the CNN-LSTM only predicts v, this layer requires inputs from two sources, v coming from the CNN-LSTM and u coming from previous predictions of the custom layer, directly bypassing the CNN-LSTM. Therefore, the previously used Sequential Application Programming Interface (API) of Tensorflow was discarded in favor of the Functional API. As the equations governing v and u require current as input, not voltage, the CNN-LSTM in this case needs to be tasked with solving for synaptic (and subsequent membrane) current. Consequently, to gauge the upper limits of this method, we administered a synaptic current waveform as input during layer evaluation.

The second approach we took for custom top layer creation involved a more conventional route, where recurrent encoder (stacked LSTM layers having first decreasing and then increasing number of units) were constructed, operating on a longer batch of CNN-LSTM predictions. Specifically, the encoder responsible for fluorescent calcium signal generation took 3 s of voltage input, while the voltage reporter encoder and decoder operated on 1024 ms of signal input.

## Computational resources

We used several different commercially available and free-to-use computational resources to demonstrate the attainableness of large network simulations using neural networks. Single-compartmental NEURON simulations were carried out on a single CPU (Intel Core i7-5557U CPU @3.1 GHz), equipped with four logical processors and two cores. Python had access to the entirety of the CPU; however, no explicit attempts were made to enable code parallelization. To test runtimes on a CPU, only a single core was used. For multicompartmental NEURON simulations, we used the publicly available National Science Foundation-funded High Performance Computing resource via the Neuroscience Gateway (*Sivagnanam et al., 2013*). This resource was only used to generate training datasets. Speed comparison using CPUs was always carried out on the aforementioned single CPU. In contrast to NEURON models, ANN calculations are designed to run on GPUs rather than CPUs. Therefore, ANN models were run on the freely accessible Google Collaboratory GPUs (NVIDIA Tesla K80), Google Collaboratory TPUs (designed for handling tensor calculations typically created by Tensorflow) or a single high-performance GPU (GeForce GTX 1080 Ti). For speed comparisons, we ran these models on a single Google Collaboratory CPU (Intel Xeon, not specified, @2.2 GHz) and the previously mentioned single CPU as well. During NEURON and ANN simulations, parallelization was only employed for Neuroscience Gateway simulations and ANN fitting.

## Statistics

Averages of multiple measurements are presented as mean ± SD. Data were statistically analyzed by ANOVA test using Origin software and custom-written Python scripts. Normality of the data was analyzed with Shapiro–Wilk test. Explained variance was quantified as 1 minus the fitting error normalized by the variance of the signal (*Ujfalussy et al., 2018*). For accuracy measurements, APs were counted within a 10 ms time window as true-positive APs. Precision and recall were calculated based on the following equations:

$$precision = \frac{TP}{TP+FP}$$

$$recall = \frac{TP}{TP+FN}$$

where FP in the false-positive rate and FN is the false-negative rate.

## Data and software availability

All codes used for simulating single- and multicompartmental NEURON models for training dataset creation, ANN benchmarking, ANN representations, and the L5 microcircuit are available on GitHub (https://github.com/ViktorJOlah/Neuro_ANN, copy archived at swh:1:rev:52616946ed-d6489a967a645bbab805577b15ad7f; *Oláh, 2022*) and Dryad.

## Acknowledgements

This work was supported by NIH grants R56-AG072473 (MJMR) and the Emory Alzheimer's Disease Research Center Grant 00100569 (MJMR) with partial support (NPP) provided by CURE Epilepsy and the National Institutes of Health K08NS105929.

## Additional information

### Funding

| Funder | Grant reference number | Author |
| --- | --- | --- |
| National Institutes of Health | R56-AG072473 | Matthew JM Rowan |
| Emory Alzheimer's Disease Research Center | 00100569 | Matthew JM Rowan |
| CURE Epilepsy and the NIH | K08NS105929 | Nigel P Pedersen |

| Funder | Grant reference number | Author |
|---|---|---|
| National Institutes of Health | RF1-AG079269 | Matthew JM Rowan |
| Emory/Georgia Tech I3 Computational and Data analysis to Advance Single Cell Biology Research Award | | Matthew JM Rowan |

The funders had no role in study design, data collection, and interpretation, or the decision to submit the work for publication.

## Author contributions

Viktor J Oláh, Nigel P Pedersen, Conceptualization, Resources, Software, Formal analysis, Funding acquisition, Investigation, Methodology, Writing – original draft, Writing – review and editing; Matthew JM Rowan, Resources, Software, Funding acquisition, Writing – original draft, Writing – review and editing

## Author ORCIDs

Viktor J Oláh (ID) http://orcid.org/0000-0002-2069-7525
Nigel P Pedersen (ID) http://orcid.org/0000-0002-8494-0635
Matthew JM Rowan (ID) http://orcid.org/0000-0003-0955-0706

## Decision letter and Author response

Decision letter https://doi.org/10.7554/eLife.79535.sa1
Author response https://doi.org/10.7554/eLife.79535.sa2

# Additional files

## Supplementary files

• Transparent reporting form

## Data availability

All code used for simulating single and multicompartmental NEURON models, ANN benchmarking, ANN representations, and the layer 5 microcircuit are available on GitHub (https://github.com/Viktor-JOlah/Neuro_ANN, copy archived at swh:1:rev:52616946edd6489a967a645bbab805577b15ad7f) and Dryad (doi: https://doi.org/10.5061/dryad.0cfxpnw60). To adhere with eLife data availability policies, we also uploaded all data points displayed in the text and figures, on Dryad (doi: https://doi.org/10.5061/dryad.0cfxpnw60) in compliance with FAIR (Findable, Accessible, Interoperable, Reusable) principles.

The following dataset was generated:

| Author(s) | Year | Dataset title | Dataset URL | Database and Identifier |
|---|---|---|---|---|
| Oláh VJ, Pedersen NP, Rowan MJM | 2022 | Ultrafast simulation of large-scale neocortical microcircuitry with biophysically realistic neurons | https://doi.org/10.5061/dryad.0cfxpnw60 | Dryad Digital Repository, 10.5061/dryad.0cfxpnw60 |

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
