## [Editor Report]

This study describes the use of artificial neural network (ANN) methods to accurately replicate the biophysical behavior of detailed single-neuron models. The method has the potential to greatly increase the speed of neuronal modeling compared to conventional differential equation-based modeling, and scales particularly well for large network models. The authors demonstrate the fidelity of their ANN model cells over a wide range of stimulus and recording conditions including electrical and optical readouts.

---

## [Decision Letter]

**Decision letter after peer review:**

Thank you for submitting your article "Ultrafast Simulation of Large-Scale Neocortical Microcircuitry with Biophysically Realistic Neurons" for consideration by *eLife*. Your article has been reviewed by 3 peer reviewers, one of whom is a member of our Board if Reviewing Editors, and the evaluation has been overseen by Joshua Gold as the Senior Editor. The following individual involved in the review of your submission has agreed to reveal their identity: Andrew P. Davison (Reviewer #3).

Essential revisions:

1. The reviewers all felt this was a potentially exciting advance for speeding up neuronal simulations.

2. Can the authors more clearly compare the accuracy of NEURON and ANN network simulations, especially as a function of the duration of simulation? Current injection comparisons would also be useful.

3. It would help to have more detail on how the approach would scale in network simulations, especially as the synaptic connectivity is increased, and more cell types are introduced.

4. The reviewers all would like to see better software documentation and tutorials for the various steps in model implementation.

5. The reviewers would like to see a clearer comparison between NEURON and the ANN on different architectures.

6. Can the authors include details on the process and computational resources required to train the ANN? One expects that this is extensively documented in the code repository, but there should be a good starting account of this in the body of the paper.

7. Can the authors place their work in a somewhat better context? The reviewers pointed out some prior work by Beniaguev et al., and would like to see more detail on how the method might handle some existing complex simulations.

*Reviewer #1 (Recommendations for the authors):*

1. The authors state that the code will be available upon publication. This precludes the ability of the reviewers to test the code, comment on its usability, and see how well it is documented. For a methods paper, this is a surprising omission and I cannot complete the review without full code availability. Ideally, this should be in an anonymous form such as uploaded to the Journal website or provided as a package for pip install.

2. Can the authors more completely document the (a) process and (b) the computational resources required for training the ANN? Ideally (a) should be packaged in a manner where the user gives the system a model specified in NEUROML or Neuron code, and it generates the ANN a few minutes later. Maybe the authors could even provide a web resource to do this. (b) is also important to know – do we need a supercomputer to train the ANN, even if it subsequently runs on a laptop? Can the authors properly benchmark this, just as they have benchmarked runtime resource requirements? For example, what does it take to train a multicompartmental model? How does it scale with the number of compartments and variety of ion channels?

Specific points

3. Figure 5 seems to show that the ANN does indeed have an internal representation of the input placement and its effect on somatic potential. It would be very useful to see if additional readouts could report dendritic potential and Ca levels. Is there a way to read out a couple of things that would be of great interest to people studying dendritic computation?

– The membrane potential at different points on the dendrites.

– The calcium levels at different points on the dendrites.

4. Can the authors provide a readout in terms of Ca fluorescent signals?

This is now one of the major ways of monitoring large numbers of neurons in vivo in networks.

5. Can the authors explain what changes in NEURON with initialization? This seems to be used as an optional step in the comparisons with the ANN neuronal mode.

*Reviewer #2 (Recommendations for the authors):*

My main comments are mostly driven by practical considerations. If one wants to use the method and the code, one would like to know the following.

– What happens if more synapses are added? For example, the L5 PC case is presented with 200 synapses. What if one needs to use 2,000 or 20,000 synapses, which is a more realistic scenario – will one need to re-train the ANN, or will it work out of the box?

– How does the model performance change with time beyond the NEURON-simulated period that ANN is trained on? I assume that after some time the voltage trace generated by the ANN will diverge from the NEURON-simulated one, especially with respect to the timing of APs. Can the authors show a figure where such divergence is characterized as a function of time? For example, if one trains the ANN for 1 second of a NEURON simulation, how well does the ANN simulation compare to the NEURON simulation at 5 seconds? How about 10 or 100 seconds?

– How well can the trained ANN mimic responses of the neuron to current injections? Current injections (e.g., with synaptic inputs blocked) are often used to probe intrinsic properties of neurons, and there's much data available from such experiments. These data provide a natural way for model builders to test how well their neuron models are working. Furthermore, realistic perturbations that one may want to model – such as optogenetic perturbations – can often be represented rather well as an injection of positive or negative current to a cell. Can the authors demonstrate that their ANN correctly reproduces a voltage response of a NEURON-simulated cell, for example, to a step current injection?

Additional comments:

– Figure 1 (and the rest of the manuscript): the variance explained for the "winning" ANN is ~50%, which doesn't sound high. However, the ANN trace looks very close to the NEURON trace. The authors may want to elaborate on the way the agreement is quantified as the variance is explained. Maybe it will help if they compute the variance explained for the voltage traces with APs clipped. Will the variance explained be much higher in that case? It might be worth reporting that along with the variance explained for the traces that include APs (as shown currently in Figure 1).

– Figure 5 – the variance explained, precision, and recall are only shown for L5 PC, but not for L2/3, L4, and L6 PC. The precision and recall for these cells are summarized in the text, combined for the 3 neurons. It would be important to show all 3 quantities individually for each neuron, just like they are shown here for the L5 PC.

– Figure 6 – As far as I can tell, these are not connected networks. Simulating 5,000 disconnected cells is very different from 5,000 highly interconnected cells, and the speed-ups can be drastically different. This is OK for the purposes of this manuscript, but the description should be clear about what's being done. The text mentions "network" everywhere in this section, including its title. The authors should change it and make it clear that simulations involve 50 or 5,000 disconnected cells. Or, if I got this wrong, and these are indeed simulations of connected networks with 50 or 5,000 cells, then please provide the description of the network connectivity, synaptic weights, etc. (In Methods, I only see the description of a 150-neuron network for Figures 7 and 8.)

– Figure 6 – also, the authors may want to say something here about the comparison of an ANN on GPU vs. NEURON on 1 CPU is not perfect. Ideally, one would run the ANN and NEURON simulations on the same parallel hardware and compare the performance as a function of the number of parallel cores used. I understand that is hard to achieve, so it is fair that the authors do not show such a comparison. However, it is instructive to consider the following thought experiment. Even if one ran the NEURON simulation of 5,000 cells on 5,000 CPUs, the performance would likely be about the same as that for one cell on one CPU. But even then, the time of the NEURON simulation would be ~185 s (for the L5 PC), whereas the time of the CNN simulation on a SINGLE GPU is ~12 s. So, the CNN is over 10 times more efficient on a single GPU than one expects NEURON to be on 5,000 CPUs.

– Simulations of the Rett syndrome model – it might be useful to give a little more detail about the network used for these simulations in the Results (otherwise one has to check Methods for all the details). The important piece to mention is that the network does not have any inhibitory cells, and instead, inhibition is provided as external inputs together with excitation. In other words, it is a feedforward inhibition model (if I understood it correctly).

– Figure 7c, parameter mapping – I assume the bar for NEURON is interpolation?

– Page 22, "which means that a complete cortical area can be simulated using only 17 ANNs" – I am not sure this is correct. The Billeh et al., model used about 100 distinct neuronal models belonging to 17 cell types. So, simulation of this model would require about 100 ANNs, rather than 17. Of course, this is still a huge improvement relative to the hundreds of thousands of neurons in the original NEURON model.

– Discussion – the authors almost do not mention the closely related work by Beniaguev et al., (Neuron, 2022), though they do cite that paper. I believe the work by Olah et al., is sufficiently different and novel, and it offers many interesting new insights as well as opportunities for computational neuroscientists who might want to use this method and code. I would suggest that the authors add a paragraph to the Discussion and describe how their work differs from Beniaguev et al., and what their unique contributions are.

– Data and software availability – the GitHub link doesn't work. I assume the authors plan to make it public upon paper publication. But it would be nice to provide the code to the reviewers, to get some idea about the completeness of the code, since it represents one of the main results of this paper. It is also important to mention that the code shared with the community should include the functions and procedures for training the ANNs. That is one of the most valuable contributions, which will be of great interest to many neuroscientists.

*Reviewer #3 (Recommendations for the authors):*

I think this study is very nice. As noted above in the Public Review, however, I think the manuscript would be greatly improved and its impact increased by (i) showing an accuracy comparison of the results obtained with NEURON and those obtained with the ANN network for the Rett syndrome circuit model, (ii) adding performance measures for the GeNN simulator, or some other simulator that is designed to run on GPUs, at least for the point neuron model.

The availability of the source code is very welcome. However, it is not well documented. The impact of this study would be increased by providing at least a README explaining the structure of the repository, and ideally by providing instructions for reproducing at least some of your results (e.g. generating the training data, training the ANNs, using the trained networks to generate predictions, etc.)

[Editors’ note: further revisions were suggested prior to acceptance, as described below.]

Thank you for resubmitting your work entitled "Ultrafast Simulation of Large-Scale Neocortical Microcircuitry with Biophysically Realistic Neurons" for further consideration by *eLife*. Your revised article has been evaluated by Joshua Gold (Senior Editor) and a Reviewing Editor.

The manuscript has been improved but there are some remaining issues that need to be addressed, as outlined below:

The authors have substantially addressed most issues raised by the reviewers.

I would like to come back to several points in the revised version where more details in the text would greatly improve the accessibility of the study.

1. One of the key earlier reviewer points has to do with scaling with connected network size, especially with very large numbers of synapses. While the authors have responded, I was not able to understand this, and hence ask for a more complete explanation in the text so that it becomes more accessible to the readers.

The authors say:

"We thank the reviewers for pointing this out. This issue is now added to the discussion. Briefly, synaptic connectivity has no impact on simulation runtimes as the matrix transformations necessary for implementing connections take place regardless of whether two given cells are connected or not. On the other hand, inclusion of additional cell types linearly increases simulation times (assuming comparable cell numbers per cell type), as every cell type warrants the execution of additional artificial neural nets every time step."

Can the authors explain this matrix transformation step and its mapping to synaptic connectivity? I did not find an explanation in either the text or the responses to the reviewers. Possibly it may help if I were to reiterate the synaptic connectivity bottleneck in conventional simulations.

2. Each individual synaptic projection introduces a distinct delay in how long it takes for the source action potential to reach the postsynaptic synapse. This delay can be up to 10ms or sometimes longer depending on axon fiber type and length. 2. Each postsynaptic synapse is usually implemented as a conductance change obeying a single or dual α function of time. such as gSyn = gPeak * 1/tp * exp(1 -t/tp) where t is time since spike arrived at synapse and tp is time of peak of synaptic conductance.

The common observation in large spiking network models is that the combination of these calculations can lead to quite large demands, including in managing the event queues to implement the synaptic delays, since the delays may be long enough to permit multiple action potentials. The synaptic dynamics of the α functions also introduce a computational cost. Since the number of synaptic connections is very large, in some large simulations the computation time is dominated by synaptic transmission.

It would be helpful if the authors can respond by addressing a few specific points, and include the information in the text.

a. Confirm and elaborate on how their method indeed accomplishes the same computations as this, both the distinct synaptic delay for each synapse, and the equivalent of α function synapses.

b. Explain how their matrix transform addresses the two computational bottlenecks that occur with the conventional simulation approach,

c. The authors on the one hand state (line 594) "the number of contact sites directly correlates with simulation runtimes and memory consumption.", and on the other they say that synaptic connectivity has no impact on simulation runtimes. Please clarify what is different here.

3. Could the authors move some further details of the ANN training into the paper? For example, I did not see the time taken to train the ANN (~24 hours from the response to reviewers) stated in the paper. It would be very helpful for people trying to implement such networks to know what to expect in terms of training resources and time, not to mention the learning curve for the researchers themselves to figure out how to do the training.

A related point: the data availability statement explains how to access the generated models. I did not see a clear mention of the code and resources used to build the ANNs from the training set.

I understand we are still in the early days of the use of this method. It took several years after the development of the underlying matrix calculation code for neuronal calculations before there were a couple of standard simulators that helped with many things from standard libraries to graphical interfaces. Nevertheless, it would be very helpful if the authors could provide a more complete indication in the paper of what it would take for users to do such model building for themselves.

---

## [Author Response]

Essential revisions:1. The reviewers all felt this was a potentially exciting advance for speeding up neuronal simulations.

We thank the reviewers for their enthusiasm.

2. Can the authors more clearly compare the accuracy of NEURON and ANN network simulations, especially as a function of the duration of simulation? Current injection comparisons would also be useful.

We thank the reviewers for these helpful points. To address these questions, we have made several changes and have now added new additional data. First, we included an additional panel detailing the absence of accumulating errors during simulations. Our results show a lack of compounding deviation from NEURON simulations both in terms of raw difference and in explained variance as well. Second, we extended the ANN model with the ability to generate firing patterns upon current injections in a separate figure. To this end, we decided to showcase a promising feature of ANNs, where preselected neuronal dynamics can be hard-coded in custom layers to aid both learning and precision. We selected the ‘Izhikevich’ formulation (Izhikevich 2003) as the basis of action potential generation and created a custom ANN layer that can handle both previously shown synaptic dynamics and current injection as well, to produce a variety of activity patterns.

3. It would help to have more detail on how the approach would scale in network simulations, especially as the synaptic connectivity is increased, and more cell types are introduced.

We thank the reviewers for pointing this out. This issue is now added to the discussion. Briefly, synaptic connectivity has no impact on simulation runtimes as the matrix transformations necessary for implementing connections take place regardless of whether two given cells are connected or not. On the other hand, inclusion of additional cell types linearly increases simulation times (assuming comparable cell numbers per cell type), as every cell type warrants the execution of additional artificial neural nets every time step.

4. The reviewers all would like to see better software documentation and tutorials for the various steps in model implementation.

We agree and have now made the code publicly available, with documentation. The code is currently available on a public Github repository (https://github.com/ViktorJOlah/Neuro_ANN), and we have uploaded it to Dryad (https://doi.org/10.5061/dryad.0cfxpnw60 , temporary link: https://datadryad.org/stash/share/keJYkykT0Vv0h6YdA1wFNMyfEBk8TUYgRk79szUJKHM).

5. The reviewers would like to see a clearer comparison between NEURON and the ANN on different architectures.

We thank the reviewers for pointing this out and made further elaborations on the subject matter. The NEURON simulation environment is designed to run on CPUs, while ANNs created in Tensorflow are most suitable for GPUs. The inherent disparity between the two computational resources raises the possibility that the main advantage of our ANN approach arises from the computational units employed. Indeed, one of the key advancement of our manuscript is the implementation of neuronal simulations on a computational resource much faster, than what has been traditionally used. However, as Tensorflow models can be run on CPUs, we purposefully compared the two simulation environments on the same resource (a single core CPU). We have shown, that even when ANN simulations were intentionally run on a slower processing units, NEURON was only faster, when single point neurons were simulated (Figure 7). This proves, that ANN simulations not only benefit from their suitability for GPUs, but are much faster on the same resource as well. The manuscript is now extended with these clarifications (page 13).

6. Can the authors include details on the process and computational resources required to train the ANN? One expects that this is extensively documented in the code repository, but there should be a good starting account of this in the body of the paper.

We agree with the reviewers, and further elaborated on the computational resources used throughout the Results section where appropriate.

7. Can the authors place their work in a somewhat better context? The reviewers pointed out some prior work by Beniaguev et al., and would like to see more detail on how the method might handle some existing complex simulations.

We agree with this point and have now included a detailed comparison of our work and the mentioned prior publication. As we mentioned in the Introduction section, it was postulated that ANNs might be used to represent neuronal activity in single cell models by Poirazi et al., in 2003. The recent publication suggested by reviewers (Beniaguev et al.,) did implement this approach in practice.

Our aim was to create a more versatile tool to serve as a viable substituent for traditional modeling systems, with substantial innovation compared with previous work. For example, our ANN model has the ability for generalization, can produce sequential output, and generates temporally accurate action potentials without external thresholding. We demonstrated that the CNN-LSTM architecture is well suited for this task. To our knowledge, this study is the first use of ANNs that can not only serve as an alternative for single cell neuronal modeling of single, biophysically and anatomically realistic neurons, but can also produce unprecedented acceleration in network simulation runtimes.

Reviewer #1 (Recommendations for the authors):1. The authors state that the code will be available upon publication. This precludes the ability of the reviewers to test the code, comment on its usability, and see how well it is documented. For a methods paper, this is a surprising omission and I cannot complete the review without full code availability. Ideally, this should be in an anonymous form such as uploaded to the Journal website or provided as a package for pip install.

We agree and have now made the code publicly available, with documentation. The code is currently available on a public Github repository (https://github.com/ViktorJOlah/Neuro_ANN), and we have uploaded it to Dryad (https://doi.org/10.5061/dryad.0cfxpnw60 , temporary link: https://datadryad.org/stash/share/keJYkykT0Vv0h6YdA1wFNMyfEBk8TUYgRk79szUJKHM) as per instructions from the *eLife* editors.

2. Can the authors more completely document the (a) process and (b) the computational resources required for training the ANN? Ideally (a) should be packaged in a manner where the user gives the system a model specified in NEUROML or Neuron code, and it generates the ANN a few minutes later. Maybe the authors could even provide a web resource to do this. (b) is also important to know – do we need a supercomputer to train the ANN, even if it subsequently runs on a laptop? Can the authors properly benchmark this, just as they have benchmarked runtime resource requirements? For example, what does it take to train a multicompartmental model? How does it scale with the number of compartments and variety of ion channels?

We thank the reviewer for pointing this out. Besides making the code publicly available, we now further elaborated on ANN training in the Results section (page 9) by including a flowchart of the training process (Figure 5 —figure supplement 2.). Although the minutes time scale is currently unfeasible for complete ANN training, as our naïve architectures take roughly one day (24 hours) to train with current methods, there are several ways to expedite this process. Notably, transfer learning allows the reuse of previously trained ANN architectures for fitting on similar cell types. In order to properly take advantage of this method, several criteria have to be fulfilled. First, the number of synaptic contacts has to be the same, as input matrices with different dimensions require different ANN architectures. Second, the spatial relationship of the contact sites must be preserved, because a key task of the ANN is to learn the rules of synaptic interplay between selected contact sites. If this relationship is disrupted, the internal logic of the ANN is useless.

Our central aim was to lower the barrier to entry for large scale simulations of realistic neurons. This barrier is often the lack of computational resources needed for executing these experiments. Therefore, we carefully considered several publicly available options, and decided that Google Colaboratory, which is a free and public resource, was ideally suited for this purpose. The most daunting task, ANN training, was carried out in Google Colaboratory. Therefore no supercomputers are needed to replicate our findings. These details are now detailed in the methods (page 27).

The field of machine learning is rapidly evolving, producing new ANN architectures and inspired solutions constantly. Therefore, the presented ANN architecture only serves to illustrate one possibility of substituting NEURON simulations with an ANN based approached. Our architecture is not yet optimized, which is illustrated by the fact that single compartmental neurons and reconstructed L5 PC ANNs have similar number of trainable parameters despite representing cells with vastly different complexities. Architecture optimization even for simple models require immense computational resources. Therefore, we did not carry out training benchmarking, as it would be performed on unoptimized architectures.

Training of biophysicaly and morphologically realistic multicompartmental was carried out in Google Collaboratory, which is a free resource. Although this process was the most computationally exhausting task included in the paper, this step didn’t require supercomputers either. However, some simplifications had to be made for this to be feasibly carried out on the mentioned resource. The main simplification is the reduction in the number of synaptic contact sites (which doesn’t limit the number of presynaptic cells; this issue is addressed in the Discussion section). The input of the ANN contained somatic membrane potentials and synaptic input vectors. Besides the first one (membrane voltage), every column in the input matrix corresponds to a synaptic contact site. Adding more synaptic contact sites would place more memory requirements on ANN training, thereby limiting the number of trainable instances, which is the most crucial aspect of ANN fitting. On the other hand, as we mentioned previously, our current ANN architectures are unoptimized, meaning that similar number of trainable parameters were fitted on passive single compartmental cells and fully reconstructed multicompartmental active cells as well, therefore the number of ion channels in the original NEURON models had no effect of ANN fitting or query.

Specific points3. Figure 5 seems to show that the ANN does indeed have an internal representation of the input placement and its effect on somatic potential. It would be very useful to see if additional readouts could report dendritic potential and Ca levels. Is there a way to read out a couple of things that would be of great interest to people studying dendritic computation?– The membrane potential at different points on the dendrites.– The calcium levels at different points on the dendrites.

We thank the reviewer for this question. As we have shown on a single compartmental cell, it is possible to read out additional features of neuronal activity from the ANN, such as sodium and potassium current fluctuations (page 8, Figure 3). However, we recognize the importance of demonstrating the ability of voltage readout from multiple subcellular segments, therefore we now have included an additional supplementary figure (Figure 5 —figure supplement 3.), demonstrating the voltage and calcium level readout from multiple neuronal compartments using an ANN representing a fully reconstructed morphologically realistic cortical neuron.

4. Can the authors provide a readout in terms of Ca fluorescent signals?This is now one of the major ways of monitoring large numbers of neurons in vivo in networks.

We thank the reviewer for this suggestion. We agree that calcium fluorescence readouts provide crucial information about network activity, and methods built around recording these signals are now cornerstones of modern neuroscience. We recognize that demonstrating calcium fluorescence output from ANNs can entice a large community of researchers to utilize ANN based network modeling. Therefore we have extended our manuscript with a supplementary figure (Figure 6 —figure supplement 1.) showcasing accurate fluorescence readout from ANNs. Briefly, fluorescent indicators give rise to a compound signal depending on internal calcium (or voltage, in case of voltage reporters) fluctuations and the dynamics of the reporter. We demonstrated previously, that CNN-LSTMs can predict ion channel dynamics and membrane voltage simultaneously, therefore the first part of the compound signal can be properly addressed. The second part, corresponding to the dynamics of the reporter, is fixed, therefore it would theoretically require no refitting in between different architectures and cell types. Therefore, we created a custom ANN encoder, which can be trained on reporter dynamics, and can subsequently be added to the trained CNN-LSTMs. This encoder can be added to every CNN-LSTM detailed in our manuscript, and requires no further training.

5. Can the authors explain what changes in NEURON with initialization? This seems to be used as an optional step in the comparisons with the ANN neuronal mode.

Initialization is an optional step in NEURON simulation that ensures that all conductances are in steady state conditions at the start of the simulation, and therefore the beginning phase of the simulation is not contaminated by initialized membrane potential dependent currents. We now have included additional clarification regarding initialization in the main text (page 13).

Reviewer #2 (Recommendations for the authors):My main comments are mostly driven by practical considerations. If one wants to use the method and the code, one would like to know the following.– What happens if more synapses are added? For example, the L5 PC case is presented with 200 synapses. What if one needs to use 2,000 or 20,000 synapses, which is a more realistic scenario – will one need to re-train the ANN, or will it work out of the box?

We thank the reviewer for the question. We agree that the 200 synapses used for L5 PC simulations do not represent all potentially active synapses, however, this number aimed to represent 200 synaptic contact sites, which can be occupied by multiple synapses at the same time. We realize that there are certain model instances necessitating a higher number of synaptic contact sites. This is feasible, but naturally results in lower simulation speeds and much higher memory consumption. To explore the details this of this issue, we now included an additional supplementary figure (Figure 5 —figure supplement 1.) demonstrating that increasing the number of synaptic contact sites yields greater accuracy compared to spatially non-discretized simulations- but only up to a certain point. Simulation accuracy was quantified by comparing voltage traces gathered from simulations with truly randomly placed synapses to simulations with the synapses placed evenly on a restricted number of synaptic contact sites. Unsurprisingly, this also causes a higher computational load and increased simulation runtimes.

The present CNN-LSTM architecture therefore allows the addition of unlimited numbers of synapses to the same synaptic contact sites, and in this case the model will work out of the box, however, if the number of synaptic contact sites is to be modified, the ANN needs to be retrained, due to the differences in graph structures.

– How does the model performance change with time beyond the NEURON-simulated period that ANN is trained on? I assume that after some time the voltage trace generated by the ANN will diverge from the NEURON-simulated one, especially with respect to the timing of APs. Can the authors show a figure where such divergence is characterized as a function of time? For example, if one trains the ANN for 1 second of a NEURON simulation, how well does the ANN simulation compare to the NEURON simulation at 5 seconds? How about 10 or 100 seconds?

We thank the reviewer for raising this question. To address the possibility of ANN divergence, we now demonstrate that prediction error is stationary and does not accumulate with time. Our simulation (25 seconds) surprisingly shows reduced prediction errors and higher explained variance over time, even when synaptic activity was increased. We attribute the lack of compounding error to the multistep training approach we took, where high emphasis was put on learning resting membrane potential as the first training step (Figure 5 —figure supplement 2.). The stimulation pattern used for predictions was completely new to the ANN, which was previously trained using different synaptic weights and input frequencies. We have extended our manuscript with these explanations and new panels (Figure 1, panel I,J,K) highlighting these findings.

– How well can the trained ANN mimic responses of the neuron to current injections? Current injections (e.g., with synaptic inputs blocked) are often used to probe intrinsic properties of neurons, and there's much data available from such experiments. These data provide a natural way for model builders to test how well their neuron models are working. Furthermore, realistic perturbations that one may want to model – such as optogenetic perturbations – can often be represented rather well as an injection of positive or negative current to a cell. Can the authors demonstrate that their ANN correctly reproduces a voltage response of a NEURON-simulated cell, for example, to a step current injection?

We thank the reviewer for raising this issue. We recognize that one of the most common ways to characterize neuronal cells is to identify its firing behavior upon current injection. This technique is widely recognized as one of the most thorough classifiers of neuronal classes besides their anatomical features, as firing properties not only hint at the repertoire of conductances shaping neuronal excitability, but also provides information about the cell’s potential in vivo functions. Therefore, we addressed this important question by creating a new, custom ANN layer capable of generating a diverse array of firing patterns (Figure 6.). This layer is a based on the ‘Izhikevich’ equations (Izhikevich 2003), extended with the option for membrane time constant modulation.

Although ANNs are considered “black boxes” which only require minimal information about the internal encoding logic, our approach here demonstrates that hard-coding different aspects of neuronal dynamics within the ANN is also possible. Furthermore, the variables of this layer are freely adjustable, therefore the experimenter gains the ability to modify the cell’s natural activity pattern instantaneously, without altering synaptic rules or having to retrain the ANN. As the Izhikevich formulation has only four variables, neuronal firing patterns can first be fitted outside of Tensorflow, with conventional fitting algorithms. We have extended our manuscript with a new figure (Figure 6) and the corresponding explanations highlighting these findings. Additional details have also been added to the methods section and this code will be made publically available as described above.

Additional comments:– Figure 1 (and the rest of the manuscript): the variance explained for the "winning" ANN is ~50%, which doesn't sound high. However, the ANN trace looks very close to the NEURON trace. The authors may want to elaborate on the way the agreement is quantified as the variance is explained. Maybe it will help if they compute the variance explained for the voltage traces with APs clipped. Will the variance explained be much higher in that case? It might be worth reporting that along with the variance explained for the traces that include APs (as shown currently in Figure 1).

We have made these changes as recommended.

– Figure 5 – the variance explained, precision, and recall are only shown for L5 PC, but not for L2/3, L4, and L6 PC. The precision and recall for these cells are summarized in the text, combined for the 3 neurons. It would be important to show all 3 quantities individually for each neuron, just like they are shown here for the L5 PC.

We have made these changes as recommended.

– Figure 6 – As far as I can tell, these are not connected networks. Simulating 5,000 disconnected cells is very different from 5,000 highly interconnected cells, and the speed-ups can be drastically different. This is OK for the purposes of this manuscript, but the description should be clear about what's being done. The text mentions "network" everywhere in this section, including its title. The authors should change it and make it clear that simulations involve 50 or 5,000 disconnected cells. Or, if I got this wrong, and these are indeed simulations of connected networks with 50 or 5,000 cells, then please provide the description of the network connectivity, synaptic weights, etc. (In Methods, I only see the description of a 150-neuron network for Figures 7 and 8.)

The cells in Figure 6. are indeed disconnected cells and we have made the recommended changes in the manuscript. We agree with the reviewer in that high levels of interconnection can cause drastically lower simulation runtimes, but importantly, only in case of NEURON simulations. Contrastingly, the speed of our ANN approach is not dependent of the level of connectivity, as connections are applied through a simple matrix transformation, which is carried out without regards to the number of connections. Consequently, the overwhelming majority of the simulation is spent on ANN evaluation, rather than implementing synaptic communication. Due to these disparate features of NEURON and ANN simulations we decided that it would be unfair to burden NEURON simulations by enforcing further runtime impeding synaptic connectivity, however these important details are now touched upon in the manuscript (page 13) to better inform the reader.

– Figure 6 – also, the authors may want to say something here about the comparison of an ANN on GPU vs. NEURON on 1 CPU is not perfect. Ideally, one would run the ANN and NEURON simulations on the same parallel hardware and compare the performance as a function of the number of parallel cores used. I understand that is hard to achieve, so it is fair that the authors do not show such a comparison. However, it is instructive to consider the following thought experiment. Even if one ran the NEURON simulation of 5,000 cells on 5,000 CPUs, the performance would likely be about the same as that for one cell on one CPU. But even then, the time of the NEURON simulation would be ~185 s (for the L5 PC), whereas the time of the CNN simulation on a SINGLE GPU is ~12 s. So, the CNN is over 10 times more efficient on a single GPU than one expects NEURON to be on 5,000 CPUs.

We thank the reviewers for pointing this out. We made additional clarifications about this issue on the main text (page 13). We agree that one of the main strengths of ANNs is their suitability for GPUs, however, we intentionally leveled the playing field by showing ANN simulations on the same single CPU as well. On Figure 6 panel b and c, the light magenta bars represent ANN simulations on the same hardware as the NEURON simulations were run on. It is important to note that the single point neuron simulation was the only case where NEURON was faster than ANNs on CPU. The advantage of our approach is twofold; first, ANNs are faster on the same hardware, and second, ANNs can take advantage of GPUs.

– Simulations of the Rett syndrome model – it might be useful to give a little more detail about the network used for these simulations in the Results (otherwise one has to check Methods for all the details). The important piece to mention is that the network does not have any inhibitory cells, and instead, inhibition is provided as external inputs together with excitation. In other words, it is a feedforward inhibition model (if I understood it correctly).

We thank the reviewers for this point, we made the proposed clarification in the main text (page 15).

– Figure 7c, parameter mapping – I assume the bar for NEURON is interpolation?

We thank the reviewers for pointing this out, we colored the bar plot similarly as in Figure 6 and added and additional label.

– Page 22, "which means that a complete cortical area can be simulated using only 17 ANNs" – I am not sure this is correct. The Billeh et al., model used about 100 distinct neuronal models belonging to 17 cell types. So, simulation of this model would require about 100 ANNs, rather than 17. Of course, this is still a huge improvement relative to the hundreds of thousands of neurons in the original NEURON model.

We agree, and we made a note of this in the main text.

– Discussion – the authors almost do not mention the closely related work by Beniaguev et al., (Neuron, 2022), though they do cite that paper. I believe the work by Olah et al., is sufficiently different and novel, and it offers many interesting new insights as well as opportunities for computational neuroscientists who might want to use this method and code. I would suggest that the authors add a paragraph to the Discussion and describe how their work differs from Beniaguev et al., and what their unique contributions are.

We thank the reviewer for pointing out the usefulness of a comparison. The work by Beniaguev et al., nicely puts the idea that single neurons can be represented by ANNs (proposed by Poirazi et al., Neuron 2003) into practice. Their ANNs are able to accurately represent membrane potential dynamics of both single compartmental and more complex, fully reconstructed cells. However, our aim was to find an ANN architecture which fulfills three main criteria. First, sequential output with small temporal increments is warranted for the dynamic implementation of synaptic connectivity. The model presented by Beniaguev et al., outputs longer predictions at once, thereby precluding the possibility of synaptic timing and weight updates. Second, a critical step in ANN performance evaluation is to demonstrate generalization, which we found to be lacking in ANNs composed of stacked convolutional layers (used by Beniaguev et al.,). These tests were carried out not only on randomly timed synaptic inputs but randomized synaptic weights as well. Third, action potential generation is not a static process in realistic neurons, as the dynamic interplay between conductance activation and inactivation can drastically alter action potential threshold. Therefore, we explored several different architectures to finally settle on CCN-LSTMs, which can learn firing, therefore making external thresholding obsolete. Due to these differences, our work does not rely on the findings presented at Benigauev et al., instead proposes an ANN architecture that can be considered a viable substitute of traditional modeling systems. We have now included a paragraph in the discussion with these details discussed in context (page 18).

– Data and software availability – the GitHub link doesn't work. I assume the authors plan to make it public upon paper publication. But it would be nice to provide the code to the reviewers, to get some idea about the completeness of the code, since it represents one of the main results of this paper. It is also important to mention that the code shared with the community should include the functions and procedures for training the ANNs. That is one of the most valuable contributions, which will be of great interest to many neuroscientists.

We agree and have now made the code publicly available, with documentation. The code is currently available on a public Github repository (https://github.com/ViktorJOlah/Neuro_ANN), and we have uploaded it to Dryad (https://doi.org/10.5061/dryad.0cfxpnw60 , temporary link: https://datadryad.org/stash/share/keJYkykT0Vv0h6YdA1wFNMyfEBk8TUYgRk79szUJKHM).

Reviewer #3 (Recommendations for the authors):I think this study is very nice. As noted above in the Public Review, however, I think the manuscript would be greatly improved and its impact increased by (i) showing an accuracy comparison of the results obtained with NEURON and those obtained with the ANN network for the Rett syndrome circuit model, (ii) adding performance measures for the GeNN simulator, or some other simulator that is designed to run on GPUs, at least for the point neuron model.

We thank the reviewer for making this point. Our aim with the Rett syndrome circuit model was to showcase the aptness of the ANN network for parameter space mapping. To compare the speed of the simulations, we created an ANN network of L5 PCs, as described in the manuscript and compared the runtime to 150 unconnected L5 PCs in NEURON. We agree that this clarification is missing from the result section and made the necessary correction (page 15).

We simulated disconnected cells for two reasons. First, while implementing synaptic connectivity in the ANN network does not constrain simulation runtime due to simple matrix transformations, connections in NEURON simulations severely impede simulation runtimes. As our ANN simulations involve dynamic alterations in the rate of connectivity, NEURON simulation runtimes can vary. Therefore, we compared the ANN simulations to the fastest possible NEURON simulations, involving zero percent connectivity. Second, the proposed simulations have been largely inspired by a previous publication (Hay, Segev 2005 Cerebral Cortex), where the authors scrutinized response properties upon altered network size and connection probabilities, which results were taken into consideration during network construction.

We agree with the reviewer in that there are simulation environments, which are specifically designed for neuronal circuit simulations, yet discussion of these were missing from our manuscript. We have now addressed the advantages of these simulators in the manuscript with the necessary citations (page 2). As stated in the manuscript, ANN simulations on a single, one compartment neuron were significantly slower compared to NEURON simulations, and we further emphasized that for simplified neurons, NEURON is one of the slower simulators. We are aware that for the GeNN simulator and other environments, dealing with ODEs are much faster, and now discuss these points in the manuscript, with the appropriate citations (page 18). We again thank the reviewer for this question.

The availability of the source code is very welcome. However, it is not well documented. The impact of this study would be increased by providing at least a README explaining the structure of the repository, and ideally by providing instructions for reproducing at least some of your results (e.g. generating the training data, training the ANNs, using the trained networks to generate predictions, etc.)

We agree and have now made the code publicly available, with documentation. The code is currently available on a public Github repository (https://github.com/ViktorJOlah/Neuro_ANN), and we have uploaded it to Dryad (https://doi.org/10.5061/dryad.0cfxpnw60 , temporary link: https://datadryad.org/stash/share/keJYkykT0Vv0h6YdA1wFNMyfEBk8TUYgRk79szUJKHM).

[Editors’ note: further revisions were suggested prior to acceptance, as described below.]

1. One of the key earlier reviewer points has to do with scaling with connected network size, especially with very large numbers of synapses. While the authors have responded, I was not able to understand this, and hence ask for a more complete explanation in the text so that it becomes more accessible to the readers.The authors say:"We thank the reviewers for pointing this out. This issue is now added to the discussion. Briefly, synaptic connectivity has no impact on simulation runtimes as the matrix transformations necessary for implementing connections take place regardless of whether two given cells are connected or not. On the other hand, inclusion of additional cell types linearly increases simulation times (assuming comparable cell numbers per cell type), as every cell type warrants the execution of additional artificial neural nets every time step."Can the authors explain this matrix transformation step and its mapping to synaptic connectivity? I did not find an explanation in either the text or the responses to the reviewers. Possibly it may help if I were to reiterate the synaptic connectivity bottleneck in conventional simulations.

We thank the editors for pointing this out. We have made clarifications in the main text (page 26) regarding the implementation of synaptic connectivity. Briefly, during the initialization of the circuit, a connectivity matrix is defined, where columns correspond to presynaptic cells, and rows correspond to postsynaptic cells. If two cells are connected, the value in the matrix is 1, otherwise it is zero. During circuit simulations, after membrane predictions took place at every time step, the program creates the next input for the next prediction cycle. This involves checking every presynaptic cell for suprathreshold activity and setting the value of the appropriate column in the input matrix to (synaptic conductance) * (synaptic connection, zero or one). Therefore, whether any two given cells are connected or not, the program checks for connections and sets the synaptic input accordingly. This means that simulation runtime doesn’t depend on synaptic connectivity. Although this is not the most efficient method for implementing connections, our aim was to increase the transparency of our code and in addition, we found that this portion of our program account for only a negligible portion of simulation runtimes.

2. Each individual synaptic projection introduces a distinct delay in how long it takes for the source action potential to reach the postsynaptic synapse. This delay can be up to 10ms or sometimes longer depending on axon fiber type and length. 2. Each postsynaptic synapse is usually implemented as a conductance change obeying a single or dual α function of time. such as gSyn = gPeak * 1/tp * exp(1 -t/tp) where t is time since spike arrived at synapse and tp is time of peak of synaptic conductance.The common observation in large spiking network models is that the combination of these calculations can lead to quite large demands, including in managing the event queues to implement the synaptic delays, since the delays may be long enough to permit multiple action potentials. The synaptic dynamics of the α functions also introduce a computational cost. Since the number of synaptic connections is very large, in some large simulations the computation time is dominated by synaptic transmission.It would be helpful if the authors can respond by addressing a few specific points, and include the information in the text.a. Confirm and elaborate on how their method indeed accomplishes the same computations as this, both the distinct synaptic delay for each synapse, and the equivalent of α function synapses.b. Explain how their matrix transform addresses the two computational bottlenecks that occur with the conventional simulation approach,c. The authors on the one hand state (line 594) "the number of contact sites directly correlates with simulation runtimes and memory consumption.", and on the other they say that synaptic connectivity has no impact on simulation runtimes. Please clarify what is different here.

We agree with the editors in that in large scale circuit simulations, handling the α function and synaptic delays comes with an immense computational burden. We have elaborated on these points in the main text (page 26). To address the specific points:

a. We did not implement synaptic delays in our code. However, as synaptic delay is assumed to be constant, at least for the purposes of large-scale simulations, implementation of this feature is straight forward. In our case, synaptic delays must be determined during circuit initialization, and appropriate buffer matrices have to be created. During the exertion of synaptic inputs upon presynaptic discharge, the input would not be immediately written to the input matrix of the subsequent prediction round, instead it would be written to the buffer matrix, which is also advanced by one time step (advancement of the matrix refers to deletion of the first row, shifting every row up, and defining the last row, based on whether new inputs arrived or not, page 26).

Our ANN approach does not use α functions. We have implemented an ANN architecture that is used for various time series forecasting problems, such as stock market predictions and weather forecasts. The CNN-LSTM architecture consists of two main functional components; convolutional layers (CNN) and long short-term memory layers (LSTM). The CNN part is responsible for pattern recognition, and the LSTM layers handle temporal aspects. If the ANN is supplemented with an appropriate amount of training data, the CNN layers can learn the distinct shapes of postsynaptic responses belonging to different synaptic contacts. Although the α functions haven’t been explicitly defined in our architecture, the ANN could accurately predict (and not calculate) the postsynaptic responses, as demonstrated by the high amount of explained variance and correlation between ground truth data and prediction in our manuscript (Figure 1,2,3,4,5,6).

b. In particular, matrix transformation only addresses issues arising from abundant connectivity. As we explained for (a.), synaptic delays were not implemented in our code, but can be circumvented by buffer matrices, and α functions are not computed explicitly, rather predicted based on phenomenological observations (page 26).

c. We thank the editors for pointing out the necessity for clarification, which we made in the main text (page 9). Throughout the manuscript, we used the number of contact sites and synaptic connectivity as not interchangeable phrases, as the two refer to differently implemented quantities. First, synaptic contact sites refer to distinguished points in the dendritic tree which onto which synaptic connections arrived in NEURON simulations. These contact sites further correspond to columns in the input matrix of ANNs, where the ANN receives an n*m matrix, and every column (except for the first one contains membrane voltages) corresponds to one of these distinguished points. These columns can be written by several presynaptic cells if they establish connection on the same synaptic contact site. Therefore, the number of synaptic contact sites is a fixed constant at ANN creation, while synaptic connectivity can be dynamically altered between simulations. As previously described, synaptic connectivity does not alter simulation runtimes, while increasing the size of the input matrix places an increased computational burden on the system.

3. Could the authors move some further details of the ANN training into the paper? For example, I did not see the time taken to train the ANN (~24 hours from the response to reviewers) stated in the paper. It would be very helpful for people trying to implement such networks to know what to expect in terms of training resources and time, not to mention the learning curve for the researchers themselves to figure out how to do the training.A related point: the data availability statement explains how to access the generated models. I did not see a clear mention of the code and resources used to build the ANNs from the training set.I understand we are still in the early days of the use of this method. It took several years after the development of the underlying matrix calculation code for neuronal calculations before there were a couple of standard simulators that helped with many things from standard libraries to graphical interfaces. Nevertheless, it would be very helpful if the authors could provide a more complete indication in the paper of what it would take for users to do such model building for themselves.

We thank the for making this point. We now included additional details in the manuscript regarding ANN training (page 22).

The code availability statement has been updated to reflect that not only the code for ANN generation, but the NEURON code used for dataset building is available as well, in the same folder.